# Endogenous protein tagging in medaka using a simplified CRISPR/Cas9 knock-in approach

**Ali Seleit\*, Alexander Aulehla, Alexandre Paix\***

Developmental Biology Unit, European Molecular Biology Laboratory, Heidelberg, Germany

**Abstract** The CRISPR/Cas9 system has been used to generate fluorescently labelled fusion proteins by homology-directed repair in a variety of species. Despite its revolutionary success, there remains an urgent need for increased simplicity and efficiency of genome editing in research organisms. Here, we establish a simplified, highly efficient, and precise strategy for CRISPR/Cas9-mediated endogenous protein tagging in medaka (*Oryzias latipes*). We use a cloning-free approach that relies on PCR-amplified donor fragments containing the fluorescent reporter sequences flanked by short homology arms (30–40 bp), a synthetic single-guide RNA and Cas9 mRNA. We generate eight novel knock-in lines with high efficiency of F0 targeting and germline transmission. Whole genome sequencing results reveal single-copy integration events only at the targeted *loci*. We provide an initial characterization of these fusion protein lines, significantly expanding the repertoire of genetic tools available in medaka. In particular, we show that the *mScarlet-pcna* line has the potential to serve as an organismal-wide label for proliferative zones and an endogenous cell cycle reporter.

**\*For correspondence:**
ali.seleit@embl.de (AS);
alexandre.paix@embl.de (AP)

**Competing interest:** The authors declare that no competing interests exist.

## Introduction

The advent of gene editing tools (*Wang et al., 2016*; *Jinek et al., 2012*; *Cong et al., 2013*) in conjunction with the expansion of sequenced genomes and engineered fluorescent proteins (*Chudakov et al., 2010*; *Shaner et al., 2013*; *Bindels et al., 2017*; *Campbell et al., 2020*) has revolutionized the ability to generate endogenous fusion protein knock-in (KI) lines in a growing number of organisms (*Paix et al., 2015*; *Paix et al., 2017a, Gratz et al., 2014*; *Kanca et al., 2019*; *Wierson et al., 2020*; *Gutierrez-Triana et al., 2018*; *Auer and Del Bene, 2014*; *Yoshimi et al., 2016*; *Yao et al., 2017*; *Cong et al., 2013*; *Dickinson et al., 2015*; *Leonetti et al., 2016*; *Wierson et al., 2019*). These molecular markers expressed at physiological levels are central to our understanding of cellular- and tissue-level dynamics during embryonic development (*Gibson et al., 2013*). To this end, researchers have utilized the *Streptococcus pyogenes* CRISPR-associated protein 9 (Cas9) and a programmed associated single-guide RNA (sgRNA) to introduce a double strand break (DSB) at a pre-defined genomic location (*Jinek et al., 2012*). Cell DNA repair mechanisms are triggered by the DSB and it has been shown that providing DNA repair donors with homology arms that match those of the targeted *locus* can lead to integration of the donor constructs containing fluorescent reporter sequences in the genome by the process of homology-directed repair (HDR) (*Danner et al., 2017*; *Jasin and Haber, 2016*; *Ceccaldi et al., 2016*; *Lisby and Rothstein, 2004*). Despite its success, HDR mediated precise single-copy KI efficiencies in vertebrate models can still be low and the process of generating KI lines remains cumbersome and time consuming. Recent reports have improved the methodology by the usage of 5′ biotinylated long homology arms that prevent concatemerization of the injected dsDNA (*Gutierrez-Triana et al., 2018*) or by linking the repair donor to the Cas9 protein (*Gu et al., 2018*; *Savic et al., 2018*; *Carlson-Stevermer et al., 2017*; *Aird et al., 2018*). In addition, repair

donors with shorter homology arms in combination with in vivo linearization of the donor plasmid have been shown to mediate efficient knock-ins in zebrafish and in mammalian cells (*Wierson et al., 2020*; *Hisano et al., 2015*; *Cristea et al., 2013*; *Yao et al., 2017*).

In this work, we establish a simplified, highly efficient, and precise strategy for CRISPR/Cas9-mediated endogenous protein tagging in medaka (*Oryzias latipes*). Our approach relies on the use of biotinylated PCR-amplified donor fragments that contain the fluorescent reporter sequences flanked by short homology arms (30–40 bp), by-passing the need for cloning or in vivo linearization. We use this approach to generate and characterize a series of novel KI lines in medaka fish (*Supplementary files 3a and 4*). By utilizing whole genome sequencing (WGS) with high coverage in conjunction with Sanger sequencing of edited *loci*, we provide strong evidence for precise single-copy integration events only at the desired *loci*. In addition to generating an endogenous ubiquitous nuclear label and novel tissue-specific reporters, the KI lines allow us to record cellular processes, such as intracellular trafficking and stress granule formation in 4D during embryonic development, significantly expanding the genetic toolkit available in medaka. Finally, we provide proof-of-principle evidence that the endogenous *mScarlet-pcna* KI we generate serves as a *bona fide* proliferative cell label and an endogenous cell cycle reporter, with broad application potential in a vertebrate model system.

## Results

### A simplified, highly efficient strategy for CRISPR/Cas9-mediated fluorescent protein knock-ins in medaka

To simplify the process of generating fluorescent protein knock-ins in medaka we utilized PCR-amplified dsDNA repair donors with short homology arms (30–40 bp). In addition, biotinylated 5′ ends were used to prevent in vivo concatemerization of DNA (*Gutierrez-Triana et al., 2018*). We used a streptavidin-tagged Cas9 (Cas9-mSA), with the goal of enhancing its binding to the biotinylated repair donor constructs (*Gu et al., 2018*). This approach by-passes the need for cloning, as the short homology arms are added during PCR amplification. Also, given a linear PCR repair donor is used, there is no need for a second gRNA for in vivo plasmid linearization (*Hoshijima et al., 2016*; *Shin et al., 2014*; *Zu et al., 2013*; *Yao et al., 2017*; *Cristea et al., 2013*; *Auer et al., 2014*; *Hisano et al., 2015*; *Li et al., 2019*; *Wierson et al., 2020*; *Kimura et al., 2014*). The three-component mix: biotinylated PCR-amplified dsDNA donors, synthetic sgRNA, and Cas9-mSA mRNA (*Supplementary file 3b-e*) was injected into one-cell-stage medaka embryos (*Figure 1* and *Figure 1—figure supplement 1*), for a detailed protocol see *Supplementary files 1 and 2*. We targeted a list of eight genes with a variety of fluorescent proteins (*Figure 1* and *Figure 1—figure supplement 1*, *Supplementary files 3a-d and 4*), both N and C terminus tags were attempted (a list of all genomic *loci* targeted can be found in *Supplementary file 3a*). Targeting efficiency in F0 ranged from 11% to 59% of embryos showing mosaic expression (*Supplementary files 3a and 4*). Control injections with the *actb* sgRNA, Cas9-mSA mRNA, and the donor eGFP construct without homology arms showed no evidence of eGFP-positive cell clones in F0 (*Supplementary file 3a*), while the same construct with homology arms resulted in 39% of surviving injected embryos showing mosaic expression of eGFP (*Supplementary files 3a and 4*). The germline transmission efficiency of fluorescent F0 fish ranged from 25% to 100% for the different targeted *loci* (*Supplementary files 3a and 4*). For F0 adults with germline transmission, the percentage of positive F1 embryos ranged between 6.6% (2/30) and 50% (25/50). Using this method, we were able to establish eight stable KI lines. Importantly, a single injection round was sufficient to generate a KI line for most targeted *loci* (7/8; *Supplementary files 3a-f and 4*). As previously reported, the *actb-eGFP* tag was embryonic lethal (*Gutierrez-Triana et al., 2018*) and we could not obtain a KI line for that *locus*. We also performed an initial comparison (using fluorescent screening in F0) between different Cas9 designs, that is, with and without mSA. Our results indicate comparable efficiencies of KI insertions in F0s, irrespective of whether a streptavidin tag was included (*Supplementary file 3g*). Combined, our results provide evidence that highly efficient targeting of endogenous *loci* with large inserts (~800 bp) is obtained in medaka using the simplified KI approach presented here (*Figure 1* and *Figure 1—figure supplement 1*). In addition to being highly efficient, this protocol is rapid and simple-to-implement, as it relies on a PCR-amplified repair construct and hence alleviates the need for any additional cloning or in vivo plasmid linearization (*Figure 1—figure supplement 1*).

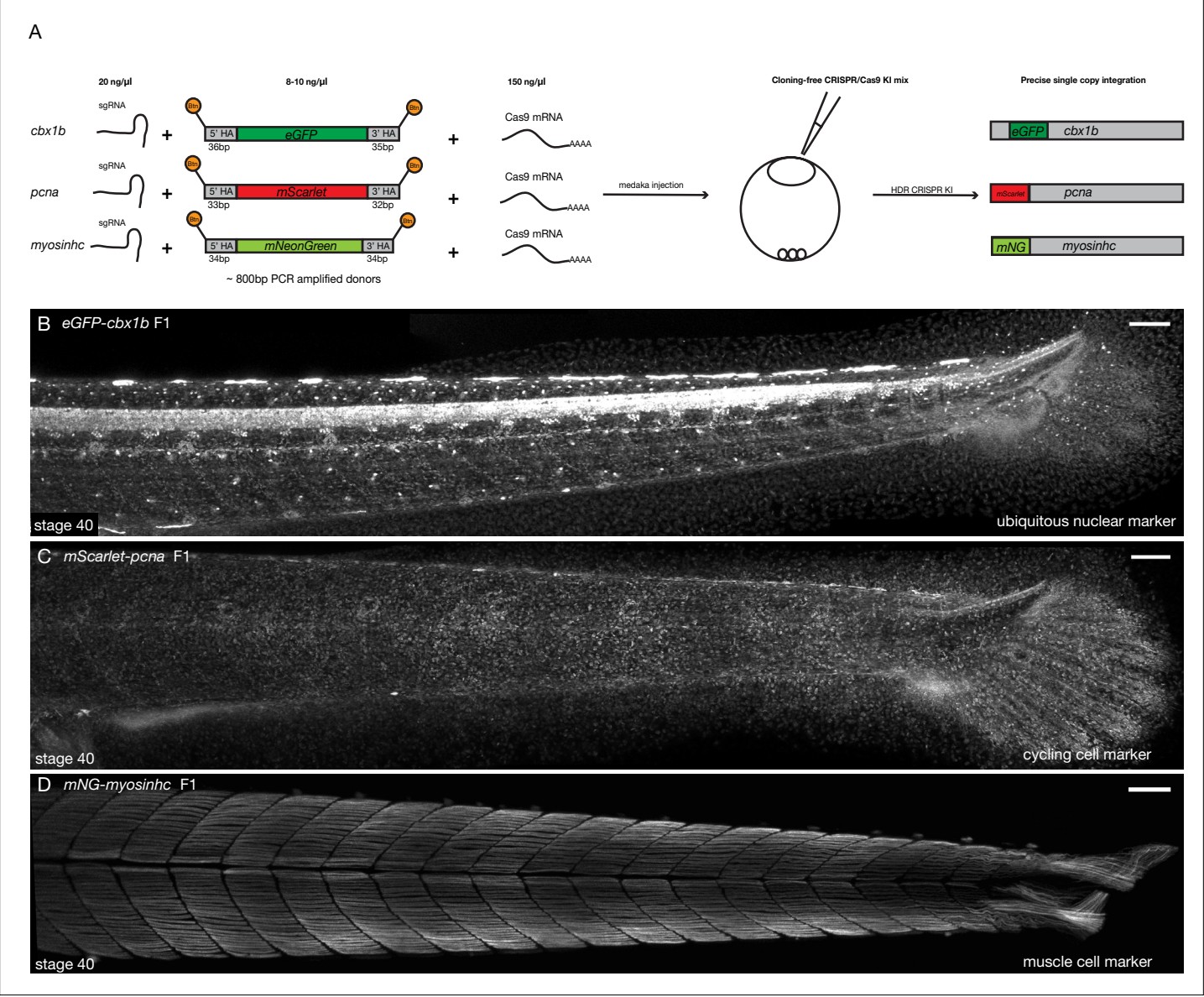

**Figure 1.** Cloning-free single-copy CRISPR/Cas9-mediated knock-in (KI) lines in medaka. (**A**) Schematic diagram of cloning-free CRISPR knock-in strategy. The injection mix consists of three components, a single-guide RNA (sgRNA) targeting the gene of interest, Cas9-mSA mRNA, and the PCR-amplified donor fragment containing short homology arms on both ends (30–40 bp) and the fluorescent protein of interest with no ATG and no stop codon. Note that the 5' ends of the PCR donor fragment are biotinylated (Btn). The mix is injected in one-cell staged medaka embryos and the injected fishes are screened for potential in-frame integrations mediated by homology-directed repair (HDR). (**B**) *eGFP-cbx1b* F1 CRISPR KI line stage 40 medaka embryos. eGFP-Cbx1b labels all nuclei and is thus an ubiquitous nuclear marker. *n* > 10 embryos. Scale bar = 100 μm (**C**) *mScarlet-pcna* F1 CRISPR KI line stage 40 medaka embryos. mScarlet-Pcna labels exclusively cycling cells. n > 10 embryos. Scale bar = 100 μm. (**D**) *mNG-myosinhc* F1 CRISPR KI line stage 40 medaka embryos. mNG-Myosinhc labels exclusively muscle cells located in the myotome tissue of medaka embryos. *n* > 10 embryos. Scale bar = 100 μm.

The online version of this article includes the following figure supplement(s) for figure 1:

**Figure supplement 1.** Schematic representation of *mNeonGreen-myosinhc* tagging strategy.

**Figure supplement 2.** Alignment of whole genome sequencing (WGS) reads to fluorescent protein sequences.

**Figure supplement 3.** Sanger sequencing of F1 lines confirms single-copy in-frame fusion proteins.

## Precise, single-copy knock-ins of fluorescent protein reporters

We next assessed the specificity and precision of the approach. It is possible that either concatemerization of inserts or off-target integrations could occur after foreign DNA delivery and CRISPR/Cas9-mediated DSBs (*Gutierrez-Triana et al., 2018*; *Doench et al., 2016*; *Fu et al., 2013*; *Paix et al., 2017a*, *Won and Dawid, 2017*; *Yan et al., 2013*; *Hackett et al., 2007*; *Wierson et al., 2020*; *Auer et al., 2014*; *Shin et al., 2014*; *Winkler et al., 1991*; *Hoshijima et al., 2016*; *Kimura et al., 2014*). To identify off-target insertions genome wide and verify single-copy integration we performed next generation WGS with high coverage (for details of WGS, see Materials and methods) on three KI lines (*Figure 1B–D*): *eGFP-cbx1b*, *mScarlet-pcna*, and *mNeonGreen-myosinhc*. For the *eGFP-cbx1b* KI line, we could only identify paired-end *eGFP* reads anchored to the endogenous *cbx1b locus* and nowhere else in the genome (*Figure 1—figure supplement 2A*). Likewise, in the *mScarlet-pcna* line, *mScarlet* reads only mapped to the endogenous *pcna locus* (*Figure 1—figure supplement 2B*). For the *mNeonGreen-myosinhc* line, *mNeonGreen* sequences mapped to the *myosinhc locus* (*Figure 1—figure supplement 2C*), but paired-end analysis yielded a second, weakly supported partial insertion of *mNeonGreen* at an intronic region in the *edf1* gene. We were not able to confirm the latter insertion by subsequent PCR and hence it remains unclear whether this a false-positive prediction or a rare insertion of very low frequency. Combined, the WGS results therefore provide strong evidence that the method we report results in single-copy insertions only at the targeted *locus*. In addition to WGS, genotyping F1 adults followed by Sanger sequencing confirmed the generation of single-copy in-frame fusion proteins in the *eGFP-cbx1b*, *mGreenLantern-cbx1b*, *mScarlet-pcna*, *mNeonGreen-myosinhc*, *cdh2-eGFP*, *mapre1b-mScarlet*, and *eGFP-rab11a* lines (*Figure 1—figure supplement 3*, *Supplementary file 3f*, and Materials and methods). Only 2/14 Sanger sequenced junctions showed evidence of imprecise repair following HDR, one of which was a partial duplication within the 5′ homology arm, 22 basepairs upstream of the start codon while the other showed a partial duplication within the 3′ homology arm four basepairs after the stop codon (for details see Materials and methods), in both cases the coding sequence of the targeted genes is unaffected. Overall, the method we present here shows high precision and specificity enabling the rapid generation of endogenously tagged alleles in a vertebrate model.

## Visualization of endogenous protein dynamics enables in vivo recording of cellular processes in medaka

As a proof of principle, we employed the simplified CRISPR/Cas9 strategy to generate a series of endogenous fusion protein KI medaka lines (*Supplementary files 3a-f and 4*; *Figures 1 and 2*, and *Figure 2—figure supplements 2–4*). Here, we provide an initial characterization of eight of these novel KI lines that are made available to the community, to label cell compartments (nucleus), cell processes (cell cycle, intra-cellular trafficking, stress granule formation), cell adhesion (adherens junctions), microtubules (plus-ends), and specific cell types (muscle cells).

### Ubiquitous nuclear marker

To generate a ubiquitously expressed nuclear label reporter line, we targeted the *cbx1b* (Chromobox protein homolog) *locus* with *eGFP* and *mGreenLantern* (*mGL*). Cbx1b is a member of the chromobox DNA-binding protein family and is a known component of heterochromatin that is expressed ubiquitously (*Lomberk et al., 2006*; *Nielsen et al., 2001*). Chromobox proteins are involved in several important functions within the nucleus, such as transcription, nuclear architecture, and DNA damage response (*Luijsterburg et al., 2009*; *Gilmore et al., 2016*). We generated two KI lines *eGFP-cbx1b* and *mGL-cbx1b* by targeting either *eGFP* or *mGL* to the N-terminus of the *cbx1b* coding sequence in medaka. The resulting lines express the fluorescent reporter in all nuclei of every tissue examined, and serve as endogenous ubiquitous nuclear labels in teleosts (*Figure 1B* and *Figure 2A*, *Figure 2—figure supplements 1 and 2*, and *Video 1*, n > 10 embryos).

### Proliferative cell marker

With the goal of generating an endogenous cell cycle reporter, we targeted the *pcna* (proliferating cell nuclear antigen) *locus* to generate a *mScarlet-pcna* fusion protein. Pcna is an essential protein regulator of DNA replication and integrity in eukaryotic cells (*Moldovan et al., 2007*; *Maga and Hubscher, 2003*; *Mailand et al., 2013*). It has been previously shown that cells that exit the cell cycle,

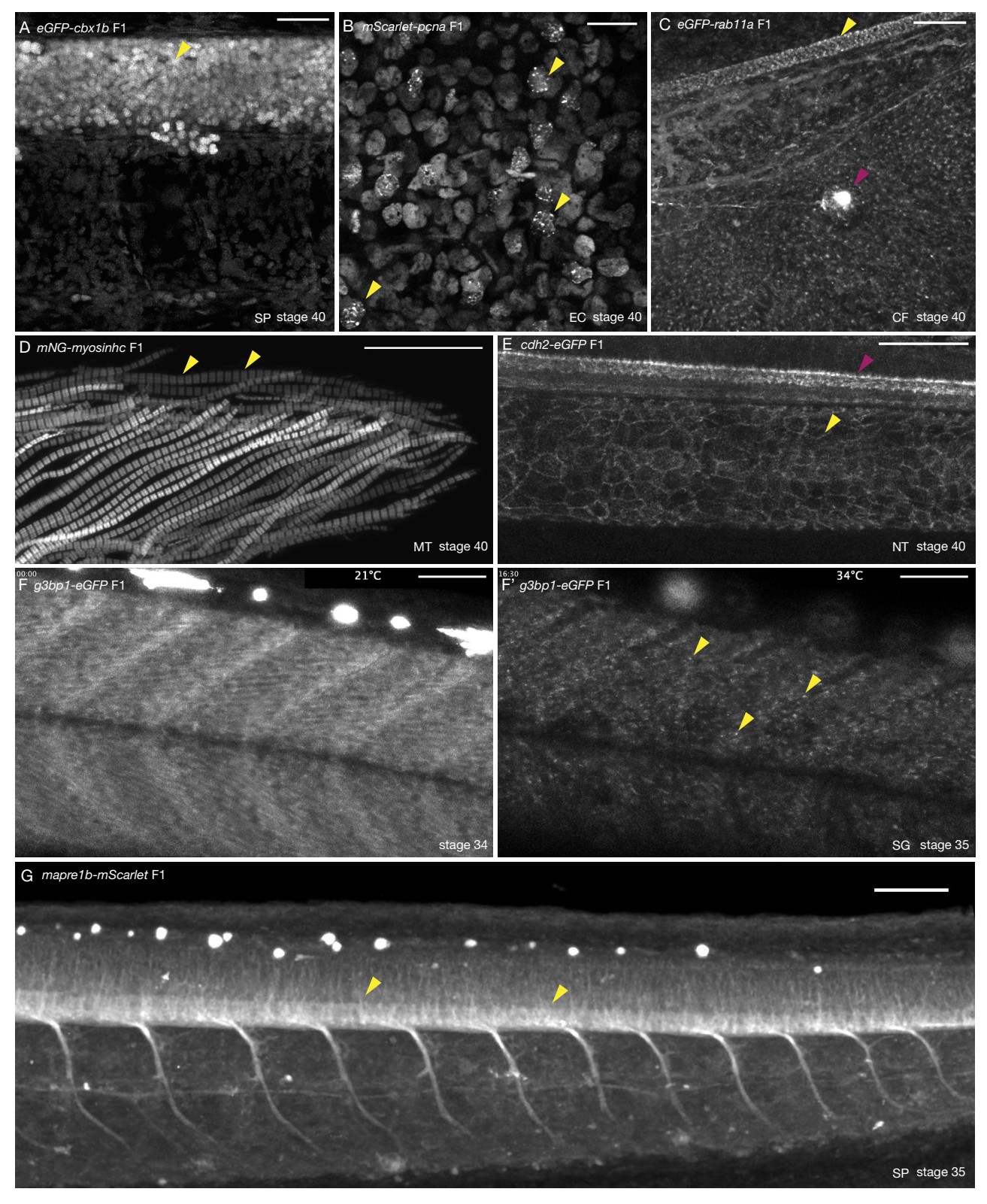

**Figure 2.** Tissue- and organelle-specific expression of seven CRISPR/Cas9 knock-in (KI) lines in medaka. (**A**) *eGFP-cbx1b* F1 stage 40 medaka embryo. eGFP-Cbx1b labels all nuclei. Nuclei in the spinal cord of medaka are highlighted (yellow arrowhead). *n* > 10 embryos. SP = spinal cord. Scale bar = 30 μm. (**B**) *mScarlet-pcna* F1 stage 40 medaka embryo. mScarlet-Pcna is localized in the nuclei of cycling cells. mScarlet-Pcna is visible in skin epithelial cell nuclei located in the mid-trunk region of a medaka embryo. The localization of Pcna as speckles within the nucleus indicates cells in S phase of the

*Figure 2 continued on next page*

*Figure 2 continued*

cell cycle (yellow arrowheads). *n* = 10 embryos. EC = epithelial cells. Scale bar = 20 µm. (**C**) *eGFP-rab11a* F1 stage 40 medaka embryo. Expression of the membrane trafficking marker eGFP-Rab11a is evident in the caudal fin region. eGFP-Rab11a is strongly expressed in the spinal cord (yellow arrowhead) and lateral line neuromasts (magenta arrowhead). *n* = 6 embryos. CF = caudal fin. Scale bar = 30 µm. (**D**) *mNG-myosinhc* F1 stage 40 medaka embryo. mNG-Myosinhc is expressed solely in muscle cells. Myofibrils containing chains of individual sarcomere can be seen (mNG-Myosinhc labels the Myosin A band inside each sarcomere, yellow arrowheads). *n* > 10 embryos. MT = myotome. Scale bar = 30 µm. (**E**) *cdh2-eGFP* F1 stage 40 medaka embryo. Cdh2-eGFP is localized at cell membranes in several tissues, including the spinal cord (magenta arrowhead) and the notochord (yellow arrowhead). *n* = 5 embryos. NT = notochord. Scale bar = 50 µm. (**F–F'**) *g3bp1-eGFP* F1 stage 34–35 medaka embryo. Time-lapse imaging of G3bp1-eGFP dynamics under normal and stress conditions. (**F**) G3bp1-eGFP localizes to the cytoplasm under physiological conditions. (**F'**) Under stress conditions (temperature shock), G3bp1-eGFP localizes to stress granules (yellow arrowheads). Time in hours. *n* = 8 embryos. SG = stress granules. Scale bar = 50 µm. (**G**) *mapre1b-mScarlet* F1 stage 35 medaka embryo. Mapre1-mScarlet is expressed in a number of tissues and cell types including epithelial cells, muscle cells, the notochord, neuromasts and is highly expressed in the spinal cord. *n* = 5 embryos. SP = spinal cord. Scale bar = 50 µm.

The online version of this article includes the following figure supplement(s) for figure 2:

**Figure supplement 1.** Ubiquitous nuclear expression of *eGFP-cbx1b*.

**Figure supplement 2.** Immunofluorescence of *eGFP-cbx1b* line confirms ubiquitous nuclear expression.

**Figure supplement 3.** Tissue-specific expression of *eGFP-rab11a* and *cdh2-eGFP* KI lines.

**Figure supplement 4.** Muscle-specific expression of *mNG-myosinhc*.

for example post-mitotic differentiated cell types, express very low levels of Pcna (*Zerjatke et al., 2017*; *Thacker et al., 2003*; *Yamaguchi et al., 1995*; *Buttitta et al., 2010*; *Alunni et al., 2010*). This has led researchers to utilize Pcna as a highly conserved marker for proliferating cells (*Zerjatke et al., 2017*; *Barr et al., 2016*; *Leonhardt et al., 2000*; *Leung et al., 2011*; *Piwko et al., 2010*; *Alunni et al., 2010*; *Thacker et al., 2003*; *Santos et al., 2015*; *Held et al., 2010*). In addition to being a specific label for cycling cells, the appearance of nuclear speckles of Pcna within the nucleus is a hallmark of cells in late S phase of the cell cycle (*Zerjatke et al., 2017*; *Barr et al., 2016*; *Leonhardt et al., 2000*; *Leung et al., 2011*; *Piwko et al., 2010*; *Santos et al., 2015*; *Held et al., 2010*). More recently, endogenously tagged Pcna has been used in mammalian cell lines to dynamically score all the different cell cycle phases (*Zerjatke et al., 2017*). We targeted the first exon of *pcna* with *mScarlet* with high efficiency (28% mosaic expression in F0s, and 50% germline transmission) and generated the *mScarlet-pcna* KI line (*Figure 1C* and *Figure 2B*). Using stage 40 medaka embryos, we detected mScarlet-Pcna-positive cells within the epidermis, specifically in supra-basal epidermal cells (*Figure 2B*, *n* = 10 embryos). A subset of these cells showed nuclear speckles of mScarlet-Pcna that likely represent replication foci and are a characteristic marker for late S phase (*Figure 2B*, yellow arrowheads). We validate the use of this line both as an organismal-wide label for proliferative zones, and an endogenous cell cycle reporter in later sections.

## Intra-cellular trafficking

To generate a reporter line allowing monitoring subcellular trafficking of endosomes and exosomes, we targeted Rab11a (Ras-Related Protein), a small GTPase and known marker of intra-cellular trafficking organelles in vertebrates (*Welz et al., 2014*; *Cullen and Steinberg, 2018*; *Stenmark, 2009*). We generated an N-terminus tagged *eGFP-rab11a* fusion protein that shows punctate intra-cellular signal most likely corresponding to trafficking organelles (*Figure 2C*, *Figure 2—figure supplement 3*, and *Videos 2–4*, *n* = 4 embryos). As a proof of principle, we detected high levels of *eGFP-rab11a* in cells of the spinal cord (*Figure 2C*, yellow arrowhead) and in neuromasts of the lateral line (*Figure 2C*, magenta arrowhead, *Figure 2—figure supplement 3*, and

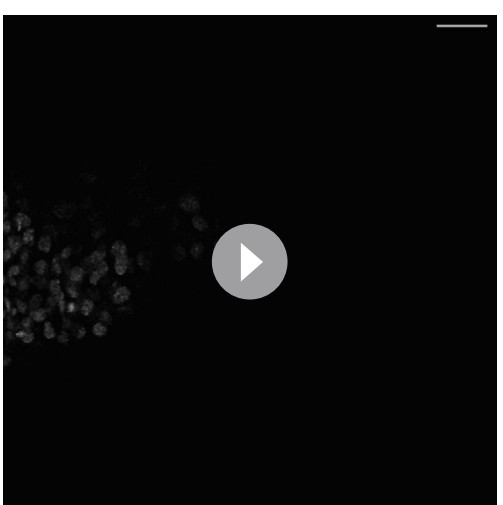

**Video 1.** Z-stack through the caudal fin region of a stage 39–40 *cbx1-eGFP* medaka embryo. eGFP-Cbx1b is expressed in all nuclei of the different cell types in the caudal fin region. *n* > 10 embryos. Scale bar = 30 µm.

https://elifesciences.org/articles/75050/figures#video1

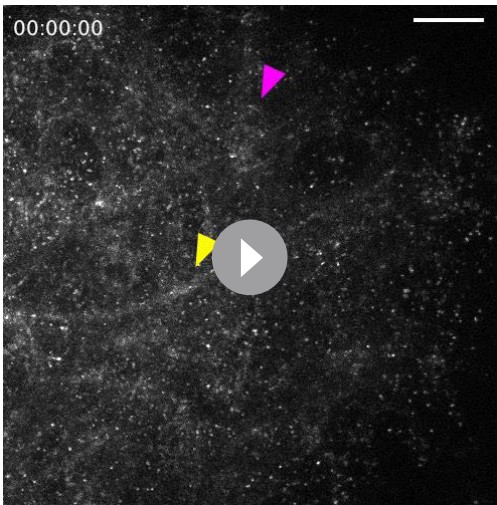

**Video 2.** Live imaging in the caudal fin region of a stage 39–40 *eGFP-rab11a* medaka embryo. eGFP-Rab11a is an intra-cellular trafficking marker and localizes to intra-cellular vesicles. Notice the dynamics of vesicle trafficking in epithelial cells, neuromasts (magenta arrowhead), and peripheral lateral line nerve (yellow arrowhead). Time in minutes. *n* = 4 embryos. Scale bar = 10 μm.

https://elifesciences.org/articles/75050/figures#video2

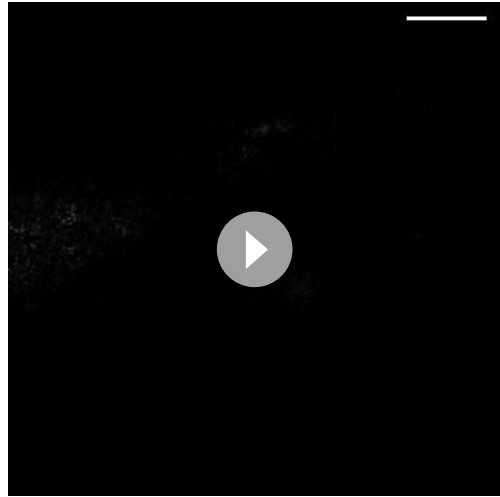

**Video 4.** Z-stack through the caudal fin region of a stage 39–40 *eGFP-rab11a* medaka embryo. eGFP-Rab11a is strongly expressed in the caudal neuromast and peripheral lateral line nerve. And is also expressed in epithelial cells, the notochord, and the spinal cord. *n* = 4 embryos. Scale bar = 30 μm.

https://elifesciences.org/articles/75050/figures#video4

*Video 4*, *n* = 6 embryos). Using the *eGFP-rab11a* KI line, we were also able to observe dynamics of what appear to be intra-cellular organelle trafficking in vivo both in individual skin epithelial cells in the mid-trunk region and in the caudal fin region of developing medaka embryos (*Videos 2 and 3*, *n* = 4 embryos) providing initial evidence of the utility of this line as a possible subcellular trafficking marker in medaka.

## Stress granule marker

We were able to generate a *g3b1-eGFP* KI line by targeting *eGFP* to the 11th exon of the medaka *g3bp1* gene. G3bp1 (GTPase activating protein SH3-domain-binding protein) is a DNA/RNA-binding protein and an initiating factor involved in stress granule formation (*Irvine et al., 2004*; *Yang et al., 2020*). Stress granules are non-membrane bound cell compartments, which form under cellular stress and accumulate non-translating mRNA and protein complexes, and play an important role

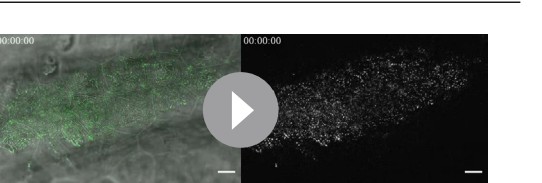

**Video 3.** Live imaging of skin epithelial cells in the mid-trunk region of a stage 39–40 *eGFP-rab11a* medaka embryo. On the left panel is a merged view of epithelial cells in bright-field and eGFP-Rab11a in green. On the right panel, eGFP-Rab11a in grey scale. eGFP-Rab11a vesicles appear as granules within the cytoplasm of epithelial cells. Time in minutes. *n* = 4 embryos. Scale bar = 10 μm.

https://elifesciences.org/articles/75050/figures#video3

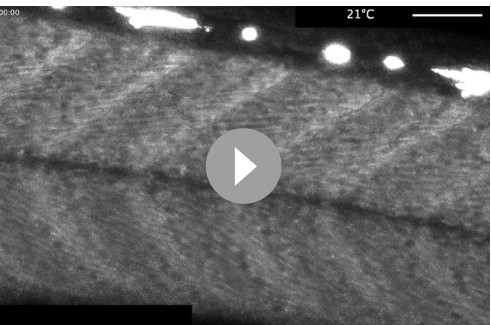

**Video 5.** Live imaging of stage 34–35 *g3bp1-eGFP* under normal conditions (temperature 21°C) reveals the cytoplasmic localization of G3bp1-eGFP in epithelial and muscle cells in the mid-trunk region of medaka embryos. Upon stress conditions (temperature shift to 34°C, 60 min after the beginning of the time-lapse), G3bp1-eGFP localization begins to shift into localized clusters of stress granule puncta. Time in hours. *n* = 8 embryos. Scale bar = 50 μm.

https://elifesciences.org/articles/75050/figures#video5

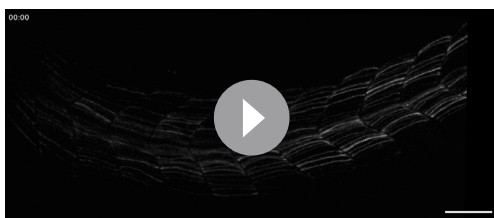

**Video 6.** Live imaging of stage 34 *mNG-myosinhc* medaka embryo during muscle formation. Muscle cell growth is driven by local buckling of individual muscle cells. Muscle growth and expression of mNG-Myosinhc does not seem to be polarized in an anterior–posterior or dorsal–ventral axis. Instead muscle cells have a heterogenous expression of mNG-Myosinhc that increases as muscle cells grow in length and mature. Time in hours. *n* = 9 embryos. Scale bar = 50 μm.
https://elifesciences.org/articles/75050/figures#video6

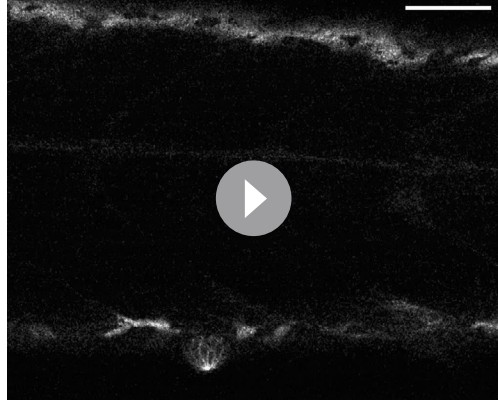

**Video 7.** Z-stack through the posterior trunk region of a stage 39–40 *cdh2-eGFP* medaka embryo. Cdh2-eGFP is expressed on the cellular membranes of neuronal tissue including neuromasts, the spinal cord, and the notochord. *n* = 3 embryos. Scale bar = 50 μm.
https://elifesciences.org/articles/75050/figures#video7

in cellular protection by regulating mRNA translation and stability (*Decker and Parker, 2012*; *Protter and Parker, 2016*). Under normal conditions G3bp1-eGFP is expressed in the cytoplasm (*Figure 2F*, *Video 5*, *n* = 8 embryos) but upon stress (temperature shock), we observe that the protein changes its localization and accumulates in cytoplasmic foci corresponding to forming stress granules (*Figure 2F'*, yellow arrowheads, *Video 5*, *n* = 8 embryos). This is consistent with previous reports showing similar changes in the localization of G3bp1 in response to stress in a number of organisms (*Guarino et al., 2019*; *Wheeler et al., 2016*; *Kuo et al., 2020*). The initial characterization of the *g3bp1-eGFP* line shows its potential to serve as a real-time in vivo reporter for the dynamics of stress granules formation in a vertebrate model.

## Muscle cell marker

To label muscle cells, we targeted muscular *myosin heavy chain* with *mNeonGreen* (*mNG*). Myosins are a highly conserved class of motor proteins implicated in actin microfilament reorganization and movement (*Sellers, 2000*; *Hartman and Spudich, 2012*). We generated an N-terminus fusion of *mNG-myosinhc* KI that exclusively labels muscle cells (*Figure 1D* and *Figure 2D* and *Figure 2—figure supplement 4*, *n* > 10 embryos). In the medaka myotome, we were able to observe *mNG-myosinhc* chains of individual sarcomeres (A-bands separated by the I-bands), indicating that tagged Myosinhc is incorporated correctly in muscle fibers (*Taylor et al., 2015*; *Loison et al., 2018*). We use this line to record the endogenous dynamics of Myosinhc during muscle growth in vivo for the first time to the best of our knowledge, in a vertebrate model (*Video 6*, *n* = 9 embryos). The *mNG-myosinhc* line therefore enables the in vivo recording of endogenous Myosinhc dynamics during myogenesis in medaka.

## Cell adhesion marker

Cadherins are a highly conserved class of transmembrane proteins that are essential components

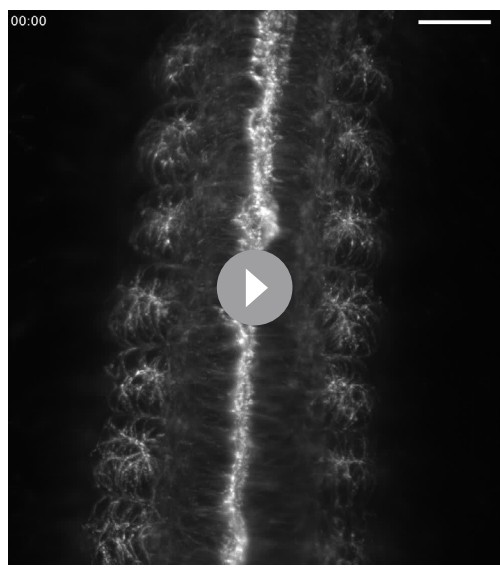

**Video 8.** Live imaging of the dorsal side of a *cdh2-eGFP* medaka embryo at the 12-somite stage reveals the endogenous dynamics of Cdh2-eGFP at high temporal resolution. Time in minutes. *n* = 2. Scale bar = 30 μm.
https://elifesciences.org/articles/75050/figures#video8

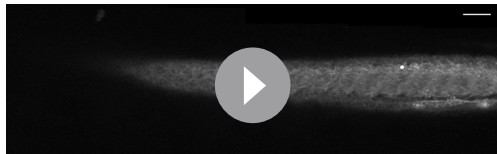

**Video 9.** Z-stack through the trunk region of a stage 35 *mapre1b-mScarlet* medaka embryo. Mapre1b-mScarlet is expressed in a number of tissues including epithelial cells, muscle cells, the notochord, neuromasts and is highly expressed in the spinal cord. *n* = 5 embryos. Scale bar = 100 μm.

https://elifesciences.org/articles/75050/figures#video9

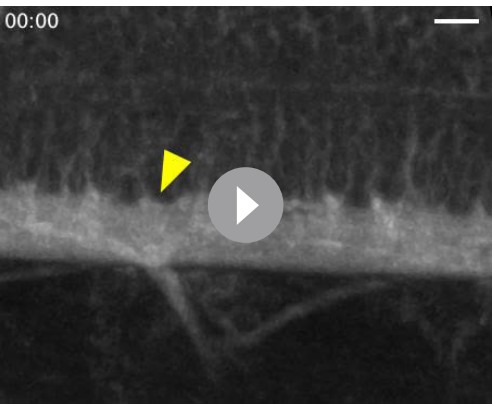

**Video 10.** Live imaging of stage 35 *mapre1b-mScarlet* medaka embryo. A close-up view on the developing spinal cord. Microtubule dynamics can be observed within the spinal cord (yellow arrowhead). Time in minutes. *n* = 5 embryos. Scale bar = 10 μm.

https://elifesciences.org/articles/75050/figures#video10

of cell–cell adhesion and are thus expressed on cellular membranes (*Leckband and de Rooij, 2014*). A large number of *cadherin* genes exist in vertebrates where they exhibit tissue-specific expression patterns and are implicated in various developmental processes (*Halbleib and Nelson, 2006*). We decided to tag the C-terminus of medaka *cadherin 2* (*cdh2, n-cadherin*) with *eGFP*. *cdh2 is* known to be expressed primarily in neuronal tissues in a number of vertebrates (*Harrington et al., 2007*; *Suzuki and Takeichi, 2008*). The *cdh2-eGFP* KI line shows cellular membrane expression in a variety of neuronal and non-neuronal tissues including the spinal cord, the eye, and the notochord (*Figure 2E*, *n* = 5 embryos) and neuromasts of the lateral line (*Figure 2—figure supplement 3*, *n* = 5 embryos, *Video 7*, *n* = 3 embryos), in addition to the developing heart (data not shown) (*Chopra et al., 2011*). The high expression of *cdh2* in both differentiated notochord cell types (*Figure 2E*, *Figure 2—figure supplement 3*) has not been previously reported in medaka but is not unexpected as this tissue experiences a high level of mechanical stress and requires strong cell–cell adhesion (*Lim et al., 2017*; *Adams et al., 1990*; *Garcia et al., 2017*; *Seleit et al., 2020*). The *cdh2-eGFP* KI can be used to study dynamics of *n-cadherin* distribution in vivo during vertebrate embryogenesis (*Video 8*, *n* = 2 embryos).

## Microtubule marker

Microtubule plus-end binding proteins are conserved regulators of microtubule dynamics, acting as a scaffold to recruit several additional proteins to ensure essential cell functions such as cell polarity, intra-cellular transport, and mitosis (*Nehlig et al., 2017*; *Tirnauer and Bierer, 2000*; *Galjart, 2010*). We successfully targeted the microtubule plus-end binding protein *mapre1b* (*eb1*), generating a C-terminal fusion protein with mScarlet. *mapre1b-mScarlet* is widely expressed in medaka embryos: epithelial cells, muscle cells, the notochord, and neuromasts all show *mapre1b-mScarlet* expression (*Figure 2G*, *Video 9*, *n* = 5 embryos), the highest level of expression occurs in the spinal cord (*Figure 2G*, yellow arrowhead). We were also able to record microtubule dynamics in the spinal cord of living embryos (*Video 10*, *n* = 5 embryos) highlighting the utility of this line for exploring the dynamics of microtubules in vivo during development.

## *mScarlet-pcna:* an organismal-wide marker for proliferative zones

We reasoned that the novel *mScarlet-pcna* line can act as an organismal-wide *bona fide* marker for the location of proliferative cells within any tissue or organ of interest. We therefore decided to generate double transgenic animals with *eGFP-cbx1b* as a ubiquitous nuclear marker and *mScarlet-pcna* as a label for cycling cells (*Figure 3*). As a proof of principle, we set out to investigate the location of proliferative zones in a number of organs and tissues in medaka. We began by assessing the position of proliferative cells in neuromast organs of the lateral line (*Seleit et al., 2017b*; *Pinto-Teixeira et al., 2015*; *Romero-Carvajal et al., 2015*). Neuromasts are small rosette shaped sensory organs located on the surface of teleost fish that sense the direction of water flow and relay the information back to the central nervous system (*Seleit et al., 2017a*; *Romero-Carvajal et al., 2015*; *Jones and*

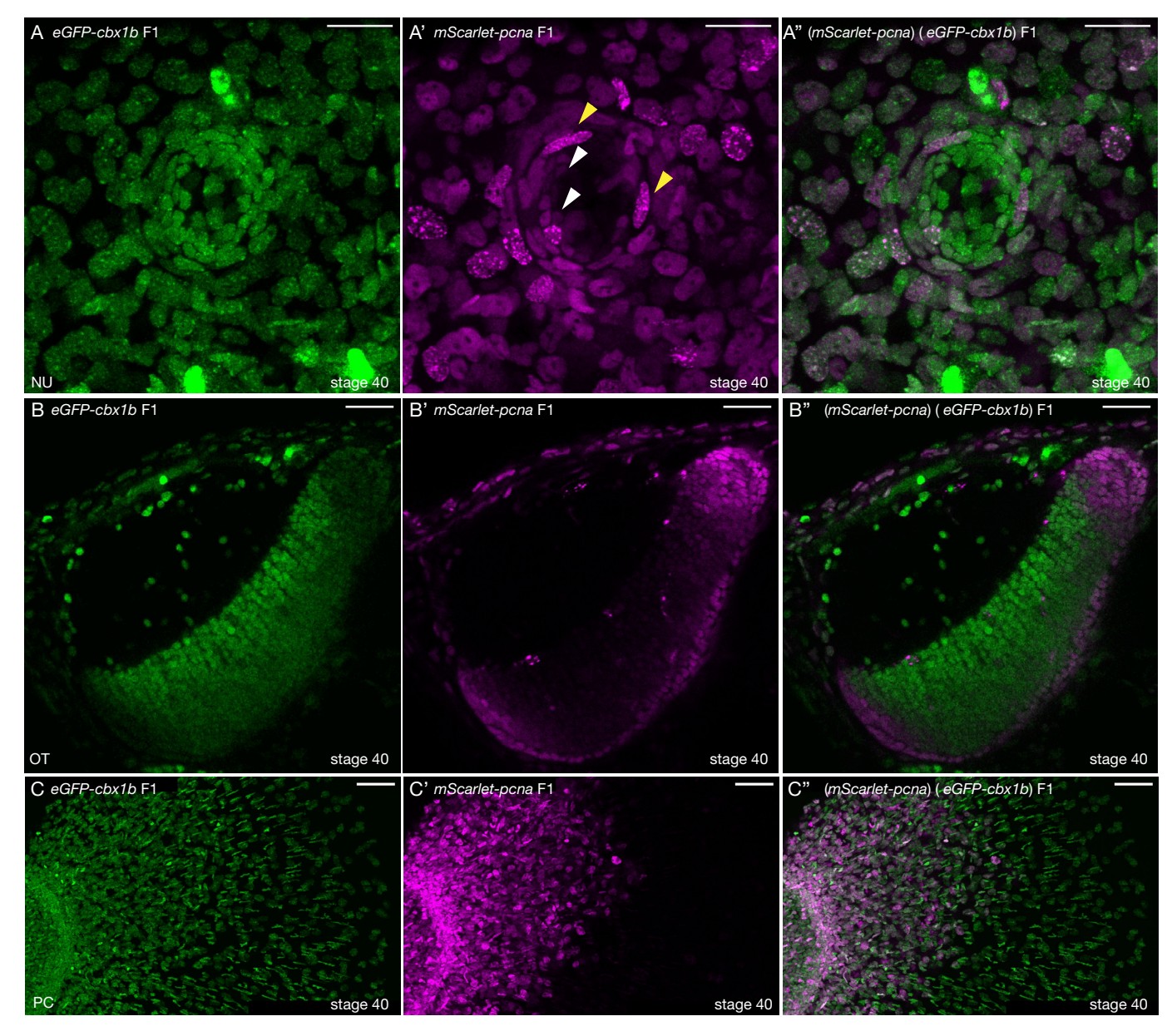

**Figure 3.** *mScarlet-pcna* line acts as an organismal-wide marker for proliferative zones. (**A–A"**) (*eGFP-cbx1b*) (*mScarlet-pcna*) double positive stage 40 medaka embryo. Maximum projection of a mature secondary neuromast (centre of image) within the lateral line system surrounded by epithelial cells, labelled by: endogenous eGFP-Cbx1b in (**A**) and endogenous mScarlet-Pcna in (**A'**). The merge is shown in (**A"**). (**A**) eGFP-Cbx1b is a ubiquitous nuclear marker and labels all cell types within a mature neuromast. Those are: hair cells (HCs) in the centre of a neuromast, surrounded by support cells (SCs), and an outer ring of mantle cells (MCs) surrounded by the elongated neuromast border cells (nBCs). (**A'**) mScarlet-Pcna labels cycling cells, which are located at the very edge of the mature neuromast organ, a proportion of MCs express mScarlet-Pcna (white arrowheads). nBCs also express mScarlet-Pcna. Speckles can be seen in several mScarlet-Pcna-positive nBC nuclei (yellow arrowheads), indicating cells in late S phase of the cell cycle. (**A"**) Merged image. *n* = 10 neuromast organs. NU = neuromast. Scale bar = 20 μm. (**B–B"**) (*eGFP-cbx1b*) (*mScarlet-pcna*) stage 40 medaka embryo. Single Z-slice showing the medaka optic tectum. (**B**) eGFP-Cbx1b is a ubiquitous nuclear marker, (**B'**) whereas mScarlet-Pcna labels a subset of cells at the outer periphery of the optic tectum, indicating the position of proliferative cells in this tissue. A graded expression of mScarlet-Pcna is observed, with more central cells in the optic tectum losing the expression of mScarlet-Pcna. (**B"**) Merged image. *n* = 4 embryos. OT = optic tectum. Scale bar = 30 μm. (**C–C"**) Maximum projection of the pectoral fin of stage 40 medaka embryos. (**C**) eGFP-Cbx1b is a ubiquitous nuclear marker, (**C'**) while a subset of cells is labelled by mScarlet-Pcna indicating the position of proliferative cells. Note the proximal to distal gradient of mScarlet-Pcna expression, with proliferative cells at the base of the fin (left) and differentiated cells at the outer edges of the fin (right) (**C"**) Merged image. *n* = 4 embryos. PC = pectoral fin. Scale bar = 50 μm.

*Figure 3 continued on next page*

*Figure 3 continued*

The online version of this article includes the following figure supplement(s) for figure 3:

**Figure supplement 1.** Proliferative cells in the anterior spinal cord.

*Corwin, 1993*; *Wada et al., 2013*). They consist of four cell types: differentiated hair cells (HCs) in the very centre, underlying support cells (SCs), a ring of mantle cells (MCs), and neuromast border cells (nBCs) (*Seleit et al., 2017b*; *Dufourcq et al., 2006*). Previous work in medaka has established MCs to be the true life-long neural stem cells within mature neuromast organs (*Seleit et al., 2017b*). While the *eGFP-cbx1b* labels all neural cells within a mature neuromast organ (HCs, SCs, and MCs) (*Figure 3A*), *mScarlet-pcna* expression matches the previously reported location of proliferative MCs (*Seleit et al., 2017b*; *Figure 3A'–A"*, white arrowhead). Neither the differentiated HCs nor the SCs directly surrounding them show evidence of Pcna expression in mature neuromast organs under homeostatic conditions in medaka (*Figure 3A–A"*, n = 10 neuromast organs). Our results validate the utility of *mScarlet-pcna* as an in vivo marker of proliferative cells. Previous work has shown that nBCs are induced to form from epithelial cells that come into contact with neuromast precursors during organ formation and that these induced cells become the stem cell niche of mature neuromast organs (*Seleit et al., 2017b*). However, an open question is whether transformed nBCs are differentiated, post-mitotic cells or whether they remain cycling. Utilizing the *mScarlet-pcna* line we were able to observe nBCs (4/42) in late S phase of the cell cycle, as evident by the presence of nuclear speckles, in mature neuromast organs (*Figure 3A'–A"*, yellow arrowheads). This provides direct evidence that nBCs retain the ability to divide and are thus not post-mitotic cells.

Next, we turned our attention to the optic tectum, which is essential for integrating visuomotor cues in all vertebrates (*Lavker and Sun, 2003*; *Alunni et al., 2010*; *Nguyen et al., 1999*). We show that proliferative cells in the optic tectum of medaka are located at the lateral, caudal, and medial edge of the tectum in a crescent-like topology (*Figure 3B–B"*, n = 4 embryos). Moreover, *mScarlet-pcna* expression is graded, with the more central cells gradually losing expression of Pcna (*Figure 3B'*). This is in line with previous histological findings using BrdU/IdU stainings in similarly staged medaka embryos (*Nguyen et al., 1999*; *Alunni et al., 2010*). We next analyzed the expression of *mScarlet-pcna* in the developing pectoral fin (*Figure 3C–C"*, n = 4 embryos). We found that cells located proximally expressed the highest levels of *mScarlet-pcna,* with *mScarlet-pcna* expression decreasing gradually along the proximo-distal axis (*Figure 3C'*). To the best of our knowledge this proliferation pattern has not been previously reported and our data provide evidence that the differentiation axis of the pectoral fin is spatially organized from proximal to distal in medaka. Lastly, we reveal that proliferative cells are present in the spinal cord of stage 40 medaka embryos, a finding that has not been previously reported, and we show that these mScarlet-Pcna-positive cells occur in clusters preferentially located on the dorsal side of the spine (*Figure 3—figure supplement 1*, n = 4 embryos). The newly developed *mScarlet-pcna* line therefore acts as a stable label of proliferative cells and as such can be used to uncover the location of proliferation zones in vivo within organs or tissues of interest in medaka.

## *mScarlet-pcna:* an endogenous cell cycle reporter

In addition to its use as a marker for cells in S phase, it has been shown that endogenously tagged Pcna can be used to determine all other cell cycle phases. This is based on the fact that both the levels and dynamic distribution of Pcna show reproducible characteristics in each phase of the cell cycle (*Held et al., 2010*; *Piwko et al., 2010*; *Santos et al., 2015*; *Zerjatke et al., 2017*; *Leonhardt et al., 2000*; *Leung et al., 2011*). To assess whether the endogenous *mScarlet-pcna* line recapitulates these known characteristic expression features during the cell cycle, we aimed to quantitatively analyze endogenous mScarlet-Pcna levels in individual cells during their cell cycle progression. To this end, we imaged skin epithelial cells located in the mid-trunk region of medaka embryos (*Figure 4*, *Figure 4—figure supplements 1 and 2*). Cells in the G1 phase of the cell cycle have been shown to decrease the levels of Pcna within the nucleus over time (*Figure 4—figure supplement 1*, *Video 11*, n = 9 epithelial cells) (*Zerjatke et al., 2017*). On the other hand, cells progressing through to S phase have been shown to increase the levels of Pcna expression within the nucleus over time (*Leonhardt et al., 2000*; *Piwko et al., 2010*; *Leung et al., 2011*; *Santos et al., 2015*; *Barr et al., 2016*; *Zerjatke et al., 2017*; *Held et al., 2010*). Indeed, we found that all tracked epithelial cells that eventually underwent

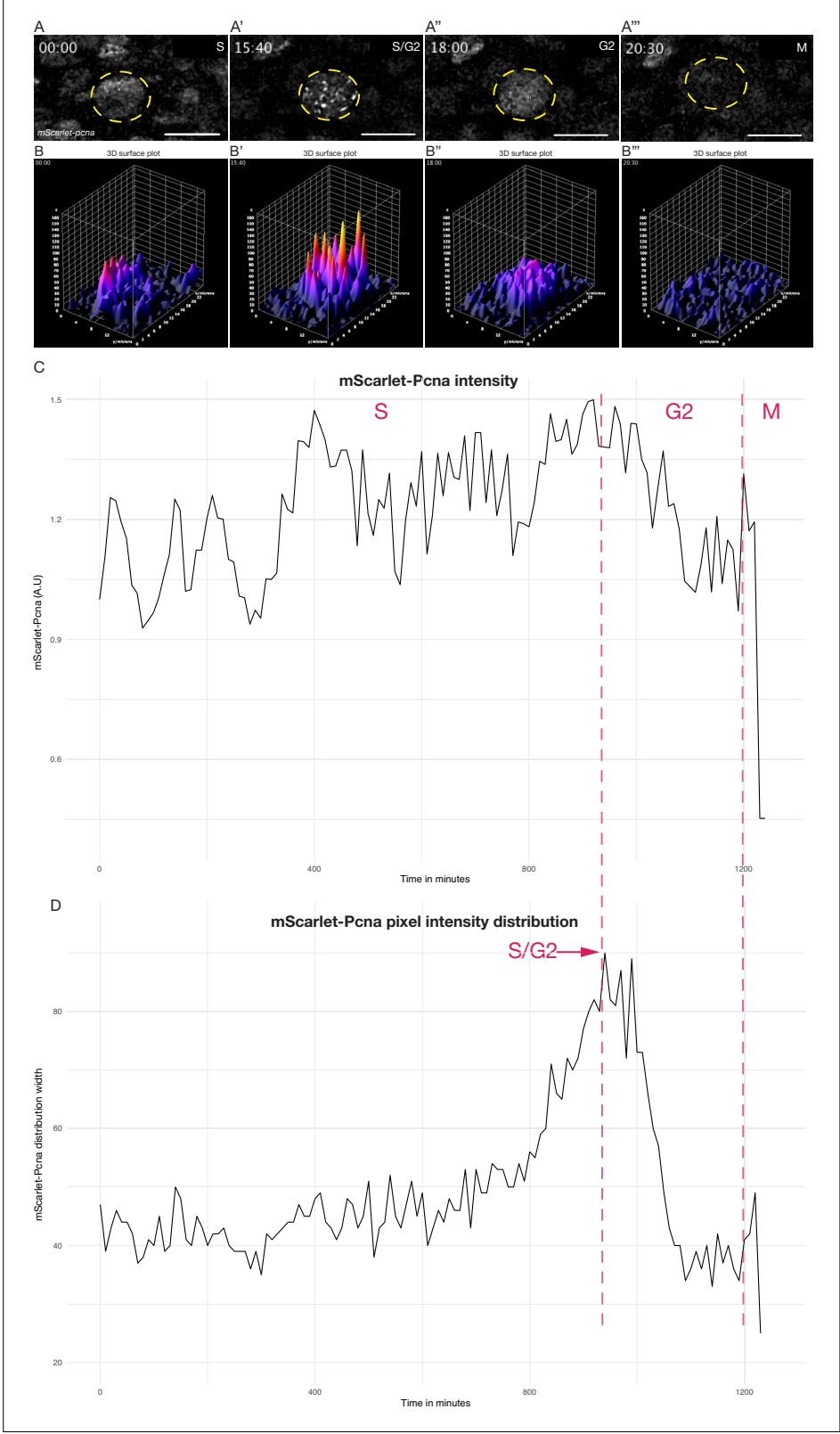

**Figure 4.** Quantitative live cell tracking of endogenous *mScarlet-Pcna* levels enables cell cycle phase classification. (**A–A'''**) Selected frames from a time-lapse imaging of a mScarlet-Pcna-positive skin epidermal cell nucleus (yellow circle) undergoing cell division. The different phases of the cell cycle are deduced from mScarlet-Pcna expression as highlighted within the panels. Late S phase can be distinguished by the presence of nuclear speckles that

*Figure 4 continued on next page*

*Figure 4 continued*

correspond to replication foci. *n* = 9 skin epithelial cells. Time in hours. (**B–B‴**) 3D surface plots of cell from (**A**). The S/G2 transition is marked as the point of peak pixel intensity distribution within the nucleus, which is reached at 15:40 hr (**B′**) and is equivalent to the largest width of mScarlet-Pcna pixel intensity distribution shown in panel D (red arrow). *n* = 9 skin epidermal cells. (**C**) Normalized mScarlet-Pcna intensity level within the nucleus from cell in (**A–A‴**) over the course of one cell division. Vertical dashed red lines demarcate the different cell cycle phases based on the intensity and distribution of mScarlet-Pcna within the nucleus. Initially, an increase of endogenous mScarlet-Pcna expression over time indicates cells in S phase of the cell cycle. M phase is characterized by a sharp drop in nuclear mScarlet-Pcna levels beginning at 1200 min. (**D**) Width of pixel intensity distribution over time on cell from (**A–A‴**). The S/G2 transition is marked as the point of peak pixel intensity distribution within the nucleus, which is reached at 940 min and is equivalent to the largest width of mScarlet-Pcna pixel intensity distribution (red arrow). *n* = 9 skin epidermal cells.

The online version of this article includes the following figure supplement(s) for figure 4:

**Figure supplement 1.** Quantification of endogenous *mScarlet-pcna* levels utilized for cell cycle phase classification.

**Figure supplement 2.** *mScarlet-pcna* histograms of pixel intensity distribution during cell cycle progression.

**Figure supplement 3.** *mScarlet-pcna* histograms of pixel intensity distribution from eight cells.

---

a cellular division showed an increase in nuclear intensity of mScarlet-Pcna prior to the appearance of nuclear speckles (***Figure 4A–B″***, ***Figure 4—figure supplement 1***, ***Video 12***, *n* = 9 epithelial cells). Nuclear speckles of Pcna mark the presence of replication foci in late S phase of the cell cycle (***Leonhardt et al., 2000***; ***Piwko et al., 2010***; ***Leung et al., 2011***; ***Barr et al., 2016***; ***Zerjatke et al., 2017***; ***Held et al., 2010***). Previous work has also shown that the S/G2 transition can be identified as the point of peak pixel intensity distribution of endogenous Pcna within the nucleus (***Zerjatke et al., 2017***). We were are able to determine the peak pixel intensity distribution within nuclei by a combination of 3D surface plots and histograms of pixel intensity distributions over time (***Figure 4B–D***, ***Figure 4—figure supplements 2 and 3***, and ***Videos 12–14***, *n* = 9 epithelial cells). Finally, onset of M phase is marked by a sharp decrease in nuclear levels of Pcna (***Zerjatke et al., 2017***; ***Leung et al., 2011***; ***Piwko et al., 2010***; ***Held et al., 2010***), which we could consistently detect in the endogenous mScarlet-Pcna inten-

sity tracks of epithelial cells undergoing division (***Figure 4A–C***, ***Figure 4—figure supplements 1 and 2***, and ***Videos 12–14***, *n* = 9 epithelial cells). We therefore provide initial evidence that the *mScarlet-pcna* line recapitulates known dynamics of Pcna within the nucleus (***Held et al., 2010***; ***Piwko et al., 2010***; ***Santos et al., 2015***; ***Zerjatke et al., 2017***; ***Barr et al., 2016***; ***Leonhardt et al.,***

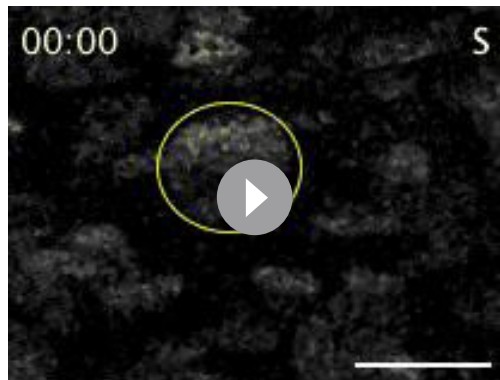

**Video 12.** Live imaging of stage 39–40 mScarlet-Pcna-positive epithelial cell nucleus undergoing cell division (shown in Figure 4). Intensity profiles were extracted from within the nucleus (yellow circle). An increase of mScarlet-Pcna levels over time is indicative of cells in the S phase of the cell cycle. The appearance of nuclear speckles is indicative of cells in late S phase. The S/G2 transition is marked as the point of peak pixel intensity distribution within the nucleus (yellow circle). The sharp drop of endogenous mScarlet-Pcna levels is indicative of cells in M phase. Time in hours. *n* = 9 mScarlet-Pcna-positive and dividing epithelial cells. Scale bar = 15 μm.
https://elifesciences.org/articles/75050/figures#video12

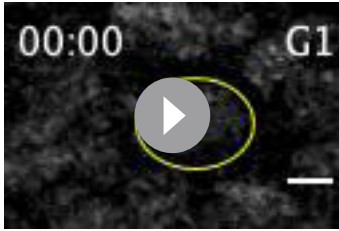

**Video 11.** Live imaging of stage 39–40 mScarlet-Pcna-positive non-dividing epithelial cell nucleus (shown in ***Figure 4—figure supplement 1***). Intensity profiles were extracted from within the nucleus (yellow circle). The cell does not divide over the course of the time-lapse. A decrease of mScarlet-Pcna levels over time is indicative of cells in the G1 phase of the cell cycle. Time in hours. *n* = 9 mScarlet-Pcna-positive non-dividing epithelial cells. Scale bar = 5 μm.
https://elifesciences.org/articles/75050/figures#video11

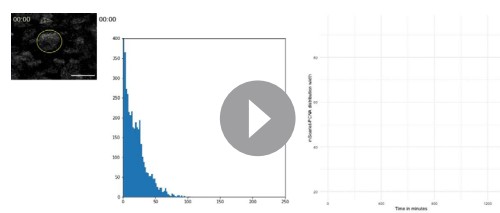

**Video 13.** Left panel: mScarlet-Pcna-positive epithelial cell nucleus undergoing cell division (shown in Figure 4). Middle panel: histogram of pixel intensity distribution of mScarlet-Pcna within the nucleus of the tracked cell. Right panel: frequency distribution width obtained from the histogram of pixel intensity distribution. Peak pixel intensity distribution is reached at 15:40 hr and is used to mark the S/G2 transition. Time in hours. *n* = 9 dividing epithelial cells. Scale bar = 15 μm.

https://elifesciences.org/articles/75050/figures#video13

2000; *Leung et al., 2011*) and that it can therefore be utilized as an endogenous 'all-in one' cell cycle reporter in vertebrates.

## Discussion

Despite the CRISPR/Cas9 system being repurposed as a broad utility genome editing tool almost a decade ago (*Jinek et al., 2012*; *Cong et al., 2013*; *Wang et al., 2016*) and despite its revolutionary impact as a method to generate knock-ins by HDR (*Danner et al., 2017*; *Jasin and Haber, 2016*; *Ceccaldi et al., 2016*; *Lisby and Rothstein, 2004*), there is still a paucity of precise, single-copy fusion protein lines in vertebrates, in general, and in teleost fish in particular. In fact, in medaka there are a total of three validated single-copy fusion protein lines by CRISPR/ Cas9 reported prior to this work (*Gutierrez-Triana et al., 2018*), and in zebrafish, a handful of lines have been reported so far (*Wierson et al., 2020*; *Hisano et al., 2015*; *Auer et al., 2014*; *Kimura et al., 2014*; *Li et al., 2019*). This underscores the complexity of generating and validating precise single-copy fusion protein KI lines in teleost models. Previous techniques to generate large KIs (such as fluorescent reporters) required the usage of plasmid vectors commonly containing long homology arms (>200 bp) (*Zu et al., 2013*; *Shin et al., 2014*; *Hoshijima et al., 2016*; *Kimura et al., 2014*; *Li et al., 2019*). Problems arising during and after injection include DNA concatemerization of the donor construct (*Gutierrez-Triana et al., 2018*; *Auer et al., 2014*; *Winkler et al., 1991*; *Hoshijima et al., 2016*; *Shin et al., 2014*), in addition to possible imprecise and off-target integration of either the fluorescent protein sequence or the plasmid backbone (*Auer et al., 2014*; *Gutierrez-Triana et al., 2018*; *Won and Dawid, 2017*; *Wierson et al., 2020*; *Shin et al., 2014*; *Hoshijima et al., 2016*; *Kimura et al., 2014*; *Li et al., 2019*; *Yan et al., 2013*; *Hackett et al., 2007*). The vast majority of reported HDR-mediated knock-ins in teleosts rely on in vivo linearization of the plasmid donors. This strategy is utilized due to the observation that, although linear dsDNA donors can drive HDR, they might be prone to degradation, concatemerization, and are generally thought to be more toxic than plasmid donors (*Auer et al., 2014*; *Cristea et al., 2013*; *Auer and Del Bene, 2014*; *Hisano et al., 2015*; *Hoshijima et al., 2016*; *Wierson et al., 2020*; *Yao et al., 2017*; *Winkler et al., 1991*). Plasmid donors therefore contain an additional guide RNA sequence to drive in vivo linearization in order to synchronize the availability of the linear DNA donor with Cas9 activity (*Auer et al., 2014*; *Cristea et al., 2013*; *Hisano et al., 2015*; *Hoshijima et al., 2016*; *Kimura et al., 2014*; *Li et al., 2019*; *Wierson et al., 2020*; *Yao et al., 2017*). We reasoned that directly injecting PCR-amplified linear DNA with short homology arms (~35 bp) could be highly effective since these donors are relatively small (~780 bp) compared to plasmids (several kbs), and therefore a small quantity of donors (~10 ng/μl) will provide a large number of molecules (~20 nM) available to engage the HDR machinery following the Cas9-induced DSB.

Building on recent improvements in CRISPR/ Cas9 KI strategies, we used 5′ biotinylated primers in order to limit in vivo concatemerization of the donor construct (*Gutierrez-Triana et al., 2018*), and synthetic sgRNAs were used to increase the efficiency of DSBs by Cas9 (*Paix et al., 2015*; *Kroll et al., 2021*; *Hoshijima et al., 2019*). In addition, we utilized a monomeric streptavidin-tagged Cas9 that has a high affinity to the biotinylated

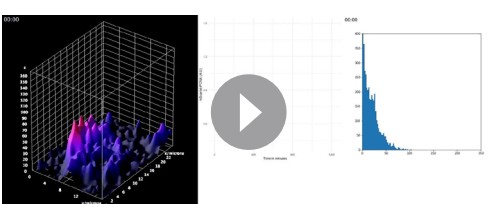

**Video 14.** Left panel: 3D surface plot of mScarlet-Pcna-positive epithelial cell nucleus undergoing cell division (from Figure 4). Middle panel: normalized endogenous mScarlet-Pcna levels of epithelial cell (from Figure 4). Right panel: histogram of pixel intensity distribution. Time in hours. *n* = 9 epithelial cells that undergo cell division.

https://elifesciences.org/articles/75050/figures#video14

donor fragments to increase targeting efficiency (*Gu et al., 2018*). While 7/8 KI lines were generated and validated using Cas9 mSA and 5′ biotinylated donor fragments, the benefit of using the mSA/Biotin system to increase targeting efficiency in F0 needs to be further evaluated: in our initial comparison, using (2xNLS) Cas9 with and without mSA, and repair donors with and without biotinylation, we have found a comparable lethality and mosaic KI efficiency rates in F0s (*Supplementary file 3g*). We also tested another Cas9 containing only one NLS and no mSA (*Gagnon et al., 2014*) and found a lower lethality rate and a lower mosaic KI efficiency in F0s compared to Cas9 with two NLSs (*Supplementary file 3g*). Irrespective of these considerations, the approach reported here is a highly efficient, precise, and scalable strategy for generating single-copy fusion proteins (*Supplementary files 3g and 4*). The fact that the repair donors are synthesized by PCR amplification eliminates the need for both cloning and a second guide RNA for in vivo linearization. Therefore, the strategy we utilize significantly simplifies the process of endogenous protein tagging in a vertebrate model.

Very recently a similar approach to the one we present here showed the potential to generate CRISPR/Cas9-mediated KI lines in zebrafish by targeting non-coding genomic regions with PCR-amplified donor constructs (*Levic et al., 2021*). Together with our present work in Medaka, this supports the notion that donors with short homology arms are sufficient to drive HDR when they are in the form of linear dsDNA. One possible explanation is that following the Cas9-induced DSB, the donor is integrated by synthesis-dependent strand annealing (SDSA). During SDSA, the 3′ ends of chromosomal DNA strands at the DSB can anneal with the repair donor and drive its replication and insertion at the site of the DSB (*Lisby and Rothstein, 2004*; *Danner et al., 2017*; *Jasin and Haber, 2016*; *Ceccaldi et al., 2016*). Short homologies (30–40 bp) appear to be sufficient to anneal with chromosomal DNA and engage the SDSA machinery (*Paix et al., 2017a*, *Paix et al., 2016*; *Grzesiuk and Carroll, 1987*).

An important aspect of any KI strategy to generate fusion proteins is its precision. The validation process of single-copy insertions is complicated in approaches that use long homology arms (>200 bps) to generate knock-ins as concatemerization (*Winkler et al., 1991*; *Gutierrez-Triana et al., 2018*) and the formation of episomes (*Aljohani et al., 2020*; *Wade-Martins et al., 1999*; *Udvadia and Linney, 2003*; *Winkler et al., 1991*; *Wierson et al., 2020*) cannot be easily ruled out. *Locus* genotyping by PCR and Sanger sequencing is difficult when using primers external to the repair donor due to the large size of the expected fragment and competition for amplification with the wildtype allele. Internal primers within the donor (junction PCR) have been used to avoid this limitation, but this can lead to PCR artefacts and crucially, it does not rule out concatemerization of the injected dsDNA (*Won and Dawid, 2017*; *Gutierrez-Triana et al., 2018*; *Auer et al., 2014*; *Hoshijima et al., 2016*; *Shin et al., 2014*). Southern blotting is considered the gold standard to assess single-copy integration (*Wierson et al., 2020*; *Gutierrez-Triana et al., 2018*; *Won and Dawid, 2017*; *Auer et al., 2014*; *Shin et al., 2014*; *Zu et al., 2013*). While it has its advantages, Southern blotting depends on experimental design (genomic DNA preparation and digestion strategy) and probe design/sensitivity, and therefore cannot exclude that part of the donor construct or part of the vector backbone integrates elsewhere in the genome. Indeed, it has been reported that plasmid donors can lead to additional unwanted insertions in the genome (*Won and Dawid, 2017*; *Auer et al., 2014*; *Hoshijima et al., 2016*; *Kimura et al., 2014*; *Li et al., 2019*; *Shin et al., 2014*). We address those issues by performing WGS with high coverage on KI lines and provide evidence that our approach yields single-copy integration only at the desired *locus*. In addition, utilizing repair donors with short homology arms on both ends (30–40 bp) simplifies the validation of the insertion by using primers that sit outside the targeting donor fragment. These external primers can then be used for genotyping of the full insertion by simple PCR followed by Sanger sequencing to know the precise nature of the edit. We show that the usage of donor fragments with short homology arms, in combination with high coverage WGS, to be important aspects in validating the precision of single-copy CRISPR/Cas9-mediated KI lines in vertebrate models.

We were able to generate eight novel endogenous protein fusion lines that significantly expand the repertoire of genetic tools to track cellular dynamics in medaka. The *eGFP-cbx1b* and *mGL-cbx1b* KI lines serve as endogenous ubiquitous nuclear markers (*Nielsen et al., 2001*; *Lomberk et al., 2006*). The generation of truly ubiquitous lines by transgene overexpression in teleost fish (*Centanin et al., 2014*; *Burket et al., 2008*) is a difficult endeavour and requires constant monitoring for variegation and silencing (*Goll et al., 2009*; *Akitake et al., 2011*; *Burket et al., 2008*; *Stuart et al., 1990*). Yet these ubiquitous fluorescent reporter lines are invaluable tools for researchers. Ubiquitous fusion

proteins expressed from the endogenous *locus* avoid potential issues with transgene overexpression and variegation. The highly conserved *cbx1b locus* could therefore provide an alternative strategy to generate faithful ubiquitous nuclear markers in other teleosts and non-model organisms. In addition, this *locus* could serve as a landing site for ubiquitous expression of genetic constructs (e.g. utilizing a 2A self-cleaving peptide) in medaka (*Li et al., 2019*; *Kim et al., 2011*). Next, we validate the use of g*3bp1-eGFP* KI as a stress granule formation marker, and utilizing 4D live imaging show the formation of stress granules in response to temperature shock in real time, as previously shown in other models using a variety of stress conditions (*Guarino et al., 2019*; *Kuo et al., 2020*; *Wheeler et al., 2016*; *Decker and Parker, 2012*; *Protter and Parker, 2016*). This line can therefore be used both as an in vivo marker of stress conditions and to study the process of stress granule formation. The *eGFP-rab11a line* serves as an intra-cellular trafficking (*Welz et al., 2014*; *Cullen and Steinberg, 2018*; *Stenmark, 2009*) marker that allows us to dynamically follow exosomes and endosomes in vivo. We report that both neuromasts and the spinal cord show substantially higher expression of *rab11a* than other tissues, the basis of this remains unclear but could indicate that these tissues exhibit higher levels of protein turnover. Despite being a highly conserved protein involved in myogenesis (*Sellers, 2000*; *Hartman and Spudich, 2012*), no endogenous KI of any myosin family member has been reported in teleosts. The *mNeonGreen-myosinhc* KI enables the detection and recording of endogenous myosin dynamics in vivo during muscle growth in a vertebrate model. We also generate *cdh2-eGFP* KI line (*Leckband and de Rooij, 2014*; *Halbleib and Nelson, 2006*) and show that it is expressed in a tissue-specific manner primarily in the spinal cord, neuromasts and the notochord. Since N-cadherin has been shown to be involved in epithelial–mesenchymal transition (EMT) (*Harrington et al., 2007*; *Suzuki and Takeichi, 2008*; *Desclozeaux et al., 2008*), this line can be used to study dynamical changes in N-cadherin distribution in vivo facilitating our understanding of EMT and other fundamental cell adhesion processes in vertebrates. Lastly, we generate and characterize the *mScarlet-pcna* KI line and discuss its usage and implications across teleosts below.

An overarching goal of developmental and stem cell biology is to discover the location of stem and progenitor cells in different organs and tissues, followed by a molecular characterization of their properties (*Rhee et al., 2006*; *Nowak et al., 2008*; *Snippert et al., 2010*; *Buczacki et al., 2013*; *Lu et al., 2012*; *Lavker and Sun, 2003*). Major advances have relied on finding resident stem cell markers that differentiates stem cells from other cell types within the same tissue, followed by BrdU/IdU staining to confirm their proliferative abilities (*Nguyen et al., 1999*; *Rhee et al., 2006*; *Nowak et al., 2008*; *Nowak and Fuchs, 2009*; *Alunni et al., 2010*; *Snippert et al., 2010*; *Lu et al., 2012*; *Buczacki et al., 2013*; *Stolper et al., 2019*; *Tsingos et al., 2019*). However, BrdU/IdU staining requires the sacrifice of the animal precluding the ability to perform 4D live imaging to analyze stem cell behaviour in vivo over time. The medaka KI line with endogenously labelled Pcna that we present here helps to circumvent this limitation. In addition, since Pcna is expressed exclusively in cycling cells (*Yamaguchi et al., 1995*; *Thacker et al., 2003*; *Buttitta et al., 2010*; *Zerjatke et al., 2017*; *Alunni et al., 2010*), it has the potential to be used to discover the location of proliferative zones in vivo within any organ or tissue of interest. We provide proof-of-principle evidence that the *mScarlet-pcna* KI line acts as a *bona fide* marker for proliferative zones in a variety of tissues in medaka fish. This line therefore represents an important new tool for stem cell research in medaka. A similar strategy could be adopted to generate endogenously tagged Pcna both in the teleost field and in other organisms.

In addition to its use as a *bona fide* marker for proliferative zones, we provide evidence that the *mScarlet-pcna* line can be used as an endogenous cell cycle reporter in medaka. It has previously been shown that both the levels and dynamic distribution of Pcna are indicative of the different cell cycle phases (*Held et al., 2010*; *Piwko et al., 2010*; *Santos et al., 2015*; *Zerjatke et al., 2017*; *Leonhardt et al., 2000*; *Barr et al., 2016*; *Leung et al., 2011*). This led researchers to successfully utilize it as an 'all-in-one' cell cycle reporter in mammalian cells (*Held et al., 2010*; *Piwko et al., 2010*; *Santos et al., 2015*; *Zerjatke et al., 2017*; *Barr et al., 2016*; *Leonhardt et al., 2000*). By quantitatively tracking endogenous Pcna levels during one cell cycle in epidermal cells of medaka fish, we were able to confirm the dynamic nature of *mScarlet-pcna* expression, which correlated with the previously described behavior of the Pcna protein within the nucleus of other vertebrates (*Held et al., 2010*; *Piwko et al., 2010*; *Santos et al., 2015*; *Zerjatke et al., 2017*; *Barr et al., 2016*; *Leonhardt et al., 2000*; *Leung et al., 2011*). As such, we provide proof-of-principle evidence that the *mScarlet-pcna* line can be successfully used as an endogenous cell cycle reporter in a teleost model. Using the

visualization of endogenous Pcna for cell cycle phase classification offers an attractive alternative to cell cycle reporters that rely on the insertion of two-colour transgenes, such as the FUCCI system (*Sugiyama et al., 2009*; *Dolfi et al., 2019*; *Araujo et al., 2016*; *Bajar et al., 2016*; *Oki et al., 2014*; *Sakaue-Sawano et al., 2008*). First, by using endogenous fusion proteins there is no requirement for overexpression of cell cycle regulators. Second, the potential issue with transgene variegation and silencing is avoided (*Akitake et al., 2011*; *Goll et al., 2009*; *Burket et al., 2008*; *Stuart et al., 1990*). Finally, utilizing a single-colour cell cycle reporter allows its simultaneous use with other fluorescent reporters during live-imaging experiments. Due to the high conservation of Pcna in eukaryotes, developing Pcna reporters in other model organisms using a similar strategy is an attractive possibility to pursue.

# Materials and methods

## Key resources table

| Reagent type (species) or resource | Designation | Source or reference | Identifiers | Additional information |
|---|---|---|---|---|
| Strain, strain background (*Oryzias latipes*) | Cab | Other | | Medaka Southern wild-type population |
| Strain, strain background (*Oryzias latipes*) | eGFP-cbx1b | This paper | | CRISPR KI line, Aulehla Lab EMBL Heidelberg |
| Strain, strain background (*Oryzias latipes*) | mGL-cbx1b | This paper | | CRISPR KI line, Aulehla Lab EMBL Heidelberg |
| Strain, strain background (*Oryzias latipes*) | cdh2-eGFP | This paper | | CRISPR KI line, Aulehla Lab EMBL Heidelberg |
| Strain, strain background (*Oryzias latipes*) | g3bp1-eGFP | This paper | | CRISPR KI line, Aulehla Lab EMBL Heidelberg |
| Strain, strain background (*Oryzias latipes*) | mapre1b-linker-3XFlag-mScarlet | This paper | | CRISPR KI line, Aulehla Lab EMBL Heidelberg |
| Strain, strain background (*Oryzias latipes*) | mNG-HAtag-Linker-myosinhc | This paper | | CRISPR KI line, Aulehla Lab EMBL Heidelberg |
| Strain, strain background (*Oryzias latipes*) | mScarlet-pcna | This paper | | CRISPR KI line, Aulehla Lab EMBL Heidelberg |
| Strain, strain background (*Oryzias latipes*) | eGFP-rab11a | This paper | | CRISPR KI line, Aulehla Lab EMBL Heidelberg |
| Strain, strain background (*Oryzias latipes*) | (eGFP-cbx1b) × (mScarlet-pcna) | This paper | | CRISPR KI line, Aulehla Lab EMBL Heidelberg |
| Antibody | Primary rabbit anti-GFP (polyclonal) | TorreyPines Biolabs | #TP401 | 1:500 |
| Antibody | Secondary goat anti-rabbit | Abcam AlexaFluor 488 | #ab150077 | 1:500 |
| Recombinant DNA reagent | Cas9-mSA plasmid | PMID:29889212 | | |
| Recombinant DNA reagent | Cas9 no-mSA plasmid | PMID:29889212 | | |
| Recombinant DNA reagent | Cas9 plasmid | PMID:24873830 | | |
| Sequence-based reagent | gRNA actb | Sigma | | Sequence (spacer) GGAUGAUGACAUUGCCGCAC |
| Sequence-based reagent | gRNA cbx1b | Sigma | | Sequence (spacer) GGAAGAUGUGGCAGAAGAAG |
| Sequence-based reagent | gRNA cdh2 | Sigma | | Sequence (spacer) GGGAGCGAUGACUAAGACAA |
| Sequence-based reagent | gRNA g3bp1 | Sigma | | Sequence (spacer) CCCCAGCGAGAGCCGCUUCU |
| Sequence-based reagent | gRNA mapre1b | Sigma | | Sequence (spacer) UCCAGAUGCUGAGGAACAGG |

*Continued on next page*

*Continued*

| Reagent type (species) or resource | Designation | Source or reference | Identifiers | Additional information |
|---|---|---|---|---|
| Sequence-based reagent | *gRNA myosinhc* | Sigma | | Sequence (spacer) CAUCUCUGCGUCAGUGCUCA |
| Sequence-based reagent | *gRNA pcna* | Sigma | | Sequence (spacer) GACCAGGCGAGCCUCAAACA |
| Sequence-based reagent | *gRNA rab11a* | Sigma | | Sequence (spacer) UCGGAUUAACGCGAGGACGA |
| Commercial assay or kit | RNAeasyMiniKit | Qiagen | #74104 | |
| Commercial assay or kit | QIAquickGel Extraction Kit | Qiagen | #28115 | |
| Commercial assay or kit | mMachineSP6 Transcription Kit | Invitrogen | #AM1340 | |
| Software, algorithm | CC-Top | PMID:25909470 | | |
| Software, algorithm | Ensembl | Public | | |
| Chemical compound, drug | Hoechst 33342 | Thermo Fischer | #H3570 | 1:500 dilution of the 10 mg/ml stock solution |
| Chemical compound, drug | Tricane | Sigma-Aldrich | #A5040-25G | |
| Chemical compound, drug | LMP Agarose | Biozyme | Plaque Agarose #840,101 | 0.6% in 1× $H_2O$ |

## Animal husbandry and ethics

Medaka (*O. latipes*, Cab strain) (*Iwamatsu, 2004*; *Naruse et al., 2004*; *Kasahara et al., 2007*) were maintained as closed stocks in a fish facility built according to the European Union animal welfare standards and all animal experiments were performed in accordance with European Union animal welfare guidelines. Animal experimentation was approved by The EMBL Institutional Animal Care and Use Committee (IACUC) project code: 20/001_HD_AA. Fishes were maintained in a constant recirculating system at 27–28°C with a 14 hr light/10 hr dark cycle.

## Cloning-free CRISPR/Cas9 knock-ins

A detailed step-by-step protocol for the cloning-free approach is provided in *Supplementary files 1 and 2*. A detailed list of all repair donors, PCR primers, fluorescent protein sequences, and sgRNAs used is provided in *Supplementary file 3a-e*. Briefly, for the preparation of Cas9-mSA mRNA: the pCS2+ Cas9 mSA plasmid was a gift from Janet Rossant (Addgene #103882, *Supplementary file 3d*; *Gu et al., 2018*). 6–8 µg of Cas9-mSA plasmid was linearized by Not1-HF restriction enzyme (NEB #R3189S). The 8.8 kb linearized fragment was cut out from a 1.5% agarose gel and DNA was extracted using QIAquick Gel Extraction Kit (Qiagen #28115). In vitro transcription was performed using mMachine SP6 Transcription Kit (Invitrogen #AM1340) following the manufacturer's guidelines. RNA cleanup was performed using RNAeasy Mini Kit (Qiagen #74104). Other Cas9 encoding plasmids used were pCS2+ Cas9 (Addgene #122948) (*Gu et al., 2018*) and pCS2-Cas9 (Addgene #47322) (*Gagnon et al., 2014*; *Supplementary file 3d,g*). sgRNAs were manually selected using previously published recommendations (*Paix et al., 2017a*; *Paix et al., 2019*; *Doench et al., 2016*; *Gagnon et al., 2014*; *Paix et al., 2017b*) and in silico validated using CCTop and CHOPCHOP (*Labun et al., 2019*; *Stemmer et al., 2015*; *Supplementary file 3e*). The genomic coordinates of all genes targeted can be found in *Supplementary file 3a*. Synthetic sgRNAs used in this study were ordered from Sigma-Aldrich (spyCas9 sgRNA, 3 nmol, HPLC purification, no modification). PCR repair donor fragments were designed and prepared as described previously (*Paix et al., 2014*; *Paix et al., 2017b*, *Paix et al., 2015*; *Paix et al., 2016*; *Paix et al., 2017a*) and a detailed protocol is provided in *Supplementary file 1*. Briefly the design includes approximately 30–40 bp of homology arms and a fluorescent protein sequence with no ATG or stop codon (*Supplementary file 3b, d*). PCR amplifications were performed using Phusion or Q5 high fidelity DNA polymerase (NEB Phusion Master Mix with HF buffer #M0531L or NEB Q5 Master Mix # M0492L). MinElute PCR Purification Kit (Qiagen #28004) was used for PCR purification. Primers were ordered from Sigma-Aldrich (25 nmol scale, desalted) and contained Biotin moiety on the 5′ ends for repair donor synthesis. A list of all primers and fluorescent

protein sequences used in this study can be found in *Supplementary file 3c, d*. The injection mix in medaka contains the sgRNA (15–20 ng/µl) + Cas9 mSA mRNA (or Cas9 mRNA without mSA) (150 ng/µl) + repair donor template (8–10 ng/µl). For injections, male and female medakas are added to the same tank and fertilized eggs collected 20 min later. The mix is injected in one-cell staged medaka embryos (*Iwamatsu, 2004*), and embryos are raised at 28°C in 1XERM (*Seleit et al., 2017a*; *Seleit et al., 2017b*; *Rembold et al., 2006*). A list of KI lines generated and maintained in this study can be found in *Supplementary file 3f*.

## Live-imaging sample preparation

Embryos were prepared for live imaging as previously described (*Seleit et al., 2017a*; *Seleit et al., 2017b*). 1× Tricaine (Sigma-Aldrich #A5040-25G) was used to anaesthetize dechorionated medaka embryos (20 mg/ml – 20× stock solution diluted in 1XERM). Anaesthetized embryos were then mounted in low melting agarose (0.6–1%) (Biozyme Plaque Agarose #840101). Imaging was done on glass-bottomed dishes (MatTek Corporation Ashland, MA, USA). For *g3bp1-eGFP* live imaging, temperature was changed from 21 to 34°C after 1 hr of imaging.

## Immunofluorescence

Immunohistochemistry was performed as previously described (*Centanin et al., 2014*). Primary rabbit anti-GFP antibody (Torrey Pines Biolabs #TP401) was used at a 1:500 dilution from the stock solution. Secondary goat anti-rabbit antibody (Abcam AlexaFluor 488 #ab150077) was used at a 1:500 dilution from the stock. Hoechst 33342 (Thermo Fischer #H3570) was used with a dilution of 1:500 of the 10 mg/ml stock solution.

## Microscopy and data analysis

For all embryo screening, a Nikon SMZ18 fluorescence stereoscope was used. All live-imaging, except for *g3bp1-eGFP* and *cdh2-eGFP embryos,* was done on a laser-scanning confocal Leica SP8 (CSU, White Laser) microscope, ×20 and ×40 objectives were used during image acquisition depending on the experimental sample. For the SP8 confocal equipped with a white laser, the laser emission was matched to the spectral properties of the fluorescent protein of interest. *g3bp1-eGFP* line live imaging was performed using a Zeiss LSM780 laser-scanning confocal with a temperature control box and an Argon laser at 488 nm, imaged through a ×20 plan apo objective (numerical aperture 0.8). For *cdh2-eGFP*, 4D live imaging was performed on a Luxendo TruLive SPIM system using a ×30 objective. Open-source standard ImageJ/Fiji software (*Schindelin et al., 2012*) was used for analysis and editing of all images post-image acquisition. Stitching was performed using standard 2D and 3D stitching plug-ins on ImageJ/Fiji. For quantitative values on endogenous mScarlet-Pcna dynamics, ROI manager in ImageJ/Fiji was used to define fluorescence intensity within the nucleus of tracked cells (yellow circle in *Figure 4* and *Videos 10 and 11*), fluorescent intensity measurements were then extracted from the time series and the data was normalized by dividing on the initial intensity value in each time-lapse movies. Data was plotted using R software. Pixel intensity distribution within nuclei were analyzed using a custom python based script (Source code file 1). Individual live-cell tracks were plotted using PlotTwist (*Goedhart, 2020*).

## Fin-clips, genotyping, and Sanger sequencing

Individual adult F1 fishes were fin clipped for genotyping PCRs. Briefly, fish were anaesthetized in 1× Tricaine solution. A small part of the caudal fin was cut by sharp scissors and placed in a 2 ml Eppendorf tube containing 50 µl of fin-clip buffer. The fishes were recovered in small beakers and were transferred back to their tanks. Eppendorf tubes were then incubated overnight at 65°C. 100 µl of $H_2O$ was then added to each tube and then the tubes were incubated for 10–15 min at 90°C. Tubes were then centrifuged for 30 min at 10,000 rpm in a standard micro-centrifuge. Supernatant was used for subsequent PCRs. Fin-clip buffer is composed of 0.4 M Tris–HCl pH 8.0, 5 mM EDTA pH 8.0, 0.15 M NaCl, 0.1% SDS in $H_2O$. 50 µl of proteinase K (20 mg/ml) was added to 1 ml fin-clip buffer before use. 2 µl of genomic DNA from fin-clips was used for genotyping PCRs. A list of all genotyping primers used in this study can be found in *Supplementary file 3c*. After PCRs the edited and wild-type amplicons were sent to Sanger sequencing (Eurofins Genomics). Sequences were analyzed using Geneious software (*Figure 1—figure supplement 3*). In-frame integrations were confirmed by

sequencing for *eGFP-cbx1b*, *mScarlet-pcna*, *mNG-myosinhc*, *eGFP-rab11a*, *mapre1b-mScarlet*, *mGL-cbx1b*, and *cdh2-eGFP*. We were able to detect an internal partial duplication of the 5′ homology arm in the *mScarlet-pcna* line that does not affect the protein coding sequence nor the 5′ extremity of the homology arm itself. Specifically, 22 basepairs upstream of the start codon of *pcna* (and within the 5′ homology arm); we detect a 21-bp partial duplication of the 5′ homology arm (CGCAACCCTCCA CAGAATAAC) and a 7-bp insertion (GGTCGAC) indicative that the repair mechanism involved can lead to errors (*Paix et al., 2017a*, *Wierson et al., 2020*). The 5′ homology junction itself is unaltered and precise. We were also able to detect a partial duplication (26 basepairs) of the 3′ homology arm in the *cdh2-eGFP* line (TTTCCTCGGTGTGGACCTTCCTACTT) that does not affect the protein coding sequence and occurs four basepairs after the stop codon.

## Whole genome sequencing

Five to ten positive F1 medaka embryos (originating from the same F0 founder) of the *eGFP-cbx1b*, *mScarlet-pcna*, and *mNeonGreen-myosinhc* lines were snap frozen in liquid nitrogen and kept at −80°C in 1.5 ml Eppendorf tubes. Genomic DNA was extracted using DNeasy Blood and Tissue Kit (Qiagen #69504) according to the manufacturer's guidelines. The libraries were prepared on a liquid handling system (Beckman i7 series) using 200 ng of sheared gDNA and 10 PCR cycles using the NEBNext Ultra II DNA Library Prep Kit for Illumina (NEB #E7645S). The DNA libraries were indexed with unique dual barcodes (8 bp long), pooled together and then sequenced using an Illumina NextSeq550 instrument with a 150 PE mid-mode in paired-end mode with a read length of 150 bp. Sequenced reads were aligned to the *O. latipes* reference genome (*Ensembl!* Assembly version ASM223467v1) using BWA mem version 0.7.17 with default settings (*Li and Durbin, 2009*). The reference genome was augmented with the known inserts for *eGFP*, *mScarlet*, and *mNeonGreen* to facilitate a direct integration discovery using standard inter-chromosomal structural variant (SV) predictions. The insert sequences are provided in *Supplementary file 3b, d*. After the genome alignment, reads were sorted and indexed using SAMtools (*Li et al., 2009*). Quality control and coverage analyses were performed using the Alfred qc subcommand (*Rausch et al., 2019*). For SV discovery, aligned reads were processed with DELLY v0.8.7 (*Rausch et al., 2012*) using paired-end mapping and split-read analysis. SVs were filtered for inter-chromosomal SVs with one breakpoint in one of the additional insert sequences (*eGFP*, *mScarlet*, and *mNeonGreen*). Plots shown in *Figure 1—figure supplement 2* were made using Integrative Genomics Viewer (IGV) (*Thorvaldsdóttir et al., 2013*). The estimated genomic coordinates for integration are: *eGFP-cbx1b* (chr19:19,074,552), *mScarlet-pcna* (chr9:6,554,003), and *mNeonGreen-myosinhc* (chr8:8,975,799). Coverage of *eGFP-cbx1b*_ gDNA1 is 20.4× and *eGFP-cbx1b*_gDNA2 is 23.6×. Coverage of *mScarlet-pcna* is 14.4×. Coverage of *mNeonGreen-myosinhc* is 14.5×. Raw sequencing data were deposited in European Nucleotide Archive (ENA) under study number ERP127162. Accession numbers are: *eGFP-cbx1b*(1) ERS5796960 (SAMEA8109891), *eGFP-cbx1b*(2) ERS5796961 (SAMEA8109892), *mScarlet-pcna* ERS5796962 (SAMEA8109893), and *mNeonGreen-myosinhc* ERS5796963 (SAMEA8109894).

## Acknowledgements

We would like to thank all members of the Aulehla lab for the fruitful discussions on the work presented here. We would like to thank Aissam Ikmi for input on the manuscript. We would like to thank Takehito Tomita for help with Python scripts. The European Molecular Biology Laboratory (EMBL-Heidelberg). Genecore is acknowledged for support in WGS data acquisition and analysis. We would like to thank Vladimir Benes and members of his team at Genecore EMBL Heidelberg for continuous help and support, Tobias Rausch for computational work on the WGS data and Mireia Osuna Lopez for help in library preparation of WGS DNA. In addition, we would like to thank all animal-care takers at EMBL Heidelberg and in particular Sabine Goergens for excellent support. We would also like to thank Addgene for access to plasmids. This work was supported by the European Molecular Biology Laboratory (EMBL-Heidelberg) and the EMBL interdisciplinary Postdoc (EIPOD4) under Marie Sklodowska-Curie Actions Cofund (grant agreement number 847543) fellowship for funding to Ali Seleit. This work also received support from the European Research Council under an ERC consolidator grant agreement no. 866537 to AA.

# Additional information

## Funding

| Funder | Grant reference number | Author |
|--------|------------------------|--------|
| H2020 European Research Council | 866537 | Alexander Aulehla<br>Ali Seleit |
| EMBL interdisciplinary Postdoc | 847543 | Ali Seleit |

The funders had no role in study design, data collection, and interpretation, or the decision to submit the work for publication.

## Author contributions

Ali Seleit, Conceptualization, Data curation, Formal analysis, Investigation, Methodology, Validation, Visualization, Writing – original draft, Writing – review and editing; Alexander Aulehla, Conceptualization, Investigation, Methodology, Validation, Visualization, Writing – original draft, Writing – review and editing; Alexandre Paix, Conceptualization, Data curation, Investigation, Methodology, Validation, Visualization, Writing – original draft, Writing – review and editing

## Author ORCIDs

Ali Seleit http://orcid.org/0000-0002-8144-2286
Alexander Aulehla http://orcid.org/0000-0003-3487-9239
Alexandre Paix http://orcid.org/0000-0002-8080-7546

## Ethics

Medaka (Oryzias latipes, Cab strain) (Iwamatsu, 2004; Naruse et al., 2004; Kasahara et al., 2007) were maintained as closed stocks in a fish facility built according to the European Union animal welfare standards and all animal experiments were performed in accordance with European Union animal welfare guidelines. Animal experimentation was approved by The EMBL Institutional Animal Care and Use Committee (IACUC) project code: 20/001_HD_AA. Fishes were maintained in a constant recirculating system at 27–28°C with a 14 hr light /10 hr dark cycle.

## Decision letter and Author response

Decision letter https://doi.org/10.7554/eLife.75050.sa1
Author response https://doi.org/10.7554/eLife.75050.sa2

# Additional files

## Supplementary files

• Supplementary file 1. Detailed protocol for cloning-free CRISPR/Cas9 Knock-In of fluorescent reporters in medaka.

• Supplementary file 2. Detailed sequence design for *mNeonGreen-HAtag-Linker-myosinhc* tagging.

• Supplementary file 3. Details of targeted *loci* and sequences for all KI lines. (a) Details of targeted *loci*. (b) PCR repair donors used in this study. (c) PCR Primers used in this study. (d) Plasmids used in this study. (e) sgRNAs used in this study. (f) Medaka lines generated and maintained in this study. (g) Comparison of Cas9 mRNAs and PCR repair donors.

• Supplementary file 4. Quantification of targeting efficiency.

• Transparent reporting form

• Source code 1. Histograms for pixel intensity distribution.

## Data availability

Sequencing data have been deposited in European Nucleotide Archive (ENA) under study number ERP127162. Accession numbers are: eGFP-cbx1b(1) ERS5796960 (SAMEA8109891), eGFP-cbx1b(2) ERS5796961 (SAMEA8109892), mScarlet-pcna ERS5796962 (SAMEA8109893) and mNeonGreen-myosinhc ERS5796963 (SAMEA8109894).

The following dataset was generated:

| Author(s) | Year | Dataset title | Dataset URL | Database and Identifier |
|---|---|---|---|---|
| Seleit A, Aulehla A, Paix A | 2021 | WGS on CRISPR mediated Knock-ins in medaka | https://www.ebi.ac.uk/ena/browser/view/PRJEB43219?show=reads | European Nucleotide Archive, PRJEB43219 |

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
