## [Decision Letter]

[Editors' note: this paper was reviewed by Review Commons.]

**Decision letter after peer review:**

Thank you for submitting your article "Endogenous protein tagging in medaka using a simplified CRISPR/Cas9 knock-in approach" for consideration by *eLife*. Your article has been reviewed by 3 peer reviewers at Review Commons, and the evaluation at *eLife* has been overseen by Didier Stainier as the Senior and Reviewing Editor.

Based on the previous reviews and the revisions, the manuscript has been improved but there are some remaining issues that need to be addressed, as outlined below:

1) Importantly, the control experiments at F0 including non-biotinylated donor templates and non-SA tagged Cas9 are now provided in the revised version (Table S7), and inconclusive results were obtained in these experiments. Comparable efficiency was obtained using both Cas9 with and without mSA, and repair donors with and without biotinylation. Moreover, likely due to the limited size of the samples, statistics analyses were not applied to these data. While these observations have only limited impact as the efficiency of the protocol is maintained independently of the biotin-streptavidin system, the manuscript needs to be clarified in terms of the mSA/Biotin dependence. Thus, this point should be openly discussed in the text to avoid misinterpretations on how the protocol works. Although this is currently mentioned in the discussion (page 10: lines 464-474), it should also be included in the Results section, and the mention of streptavidin tagged Cas9 should be removed from the abstract, and the recommendation to use streptavidin tagged Cas9 should be removed from the methods section. Moreover, the central scheme of the protocol in Figure 1A seems misleading in light of the new controls, and it should be modified to be in agreement with the fact that "the utility of the mSA/Biotin system to increase targeting efficiency in F0 needs to be further evaluated".

---

## [Author Response]

Based on the previous reviews and the revisions, the manuscript has been improved but there are some remaining issues that need to be addressed, as outlined below:1) Importantly, the control experiments at F0 including non-biotinylated donor templates and non-SA tagged Cas9 are now provided in the revised version (Table S7), and inconclusive results were obtained in these experiments. Comparable efficiency was obtained using both Cas9 with and without mSA, and repair donors with and without biotinylation. Moreover, likely due to the limited size of the samples, statistics analyses were not applied to these data. While these observations have only limited impact as the efficiency of the protocol is maintained independently of the biotin-streptavidin system, the manuscript needs to be clarified in terms of the mSA/Biotin dependence. Thus, this point should be openly discussed in the text to avoid misinterpretations on how the protocol works. Although this is currently mentioned in the discussion (page 10: lines 464-474), it should also be included in the Results section, and the mention of streptavidin tagged Cas9 should be removed from the abstract, and the recommendation to use streptavidin tagged Cas9 should be removed from the methods section. Moreover, the central scheme of the protocol in Figure 1A seems misleading in light of the new controls, and it should be modified to be in agreement with the fact that "the utility of the mSA/Biotin system to increase targeting efficiency in F0 needs to be further evaluated".

We thank the senior editor for his comments. We have followed all suggestions.

We have replaced Cas9-mSA in the abstract with Cas9 mRNA (line 26).

We have integrated the results from Supp Table 7 into the Results section as suggested (line 128-131).

We have updated the scheme in Figure 1 and removed the reference to Cas9-mSA, instead replacing it with Cas9 mRNA.

We have updated the methods section to reflect our usage of other Cas9s (without mSA). We have also removed the recommendation of using the Cas9-mSA from the methods section in Supplementary file 1 (the protocol) under the section “Cas9 mRNA in vitro transcription”.

[Editors' note: we include below the reviews that the authors received from Review Commons, along with the authors’ responses.]

Reviewer #1 (Evidence, reproducibility and clarity (Required)):This is an excellent follow up study that uses new approaches to deliver cargos for homology-directed repair in Medaka. The goal of the study was to improve the rates of targeted knock-in of DNA to create fluorescently tagged alleles to follow protein localization and gene expression. The authors nicely characterized a scarlet-pcna line that can be used to follow cell division. This should be widely requested and cited. They also created eGFP-cbx1b to follow nuclei and mNeongreen-myosin-hc line to visualize muscle, an eGFP-rab11a line to visualize vesicle transport, a g3b1-eGFP to follow stress granules, and a cdh2-eGFP line to follow cellular junctions. The real contribution here might be the method. For ease of production, the authors use PCR templates that have biotin at the ends to block concatemer formation with only short regions of homology, around 30-40 bases. They describe precise junctions following targeting, minimal off-target effects, and high rates of integration (11-59% F0 showing mosaic expression, 25-100% germline transmission, 6-50% positive F1). While the use of short homology, biotin modification of DNA ends and the use of a Cas9 fused to monomeric streptavidin (Cas9-mSA) have all been previously described, their use together to obtain high rates of targeted integration is impressive. The ability to simply add your homology arms by PCR simplifies and streamlines the production of templates. I look forward to trying this in my own laboratory.

We thank the Reviewer for his/her comment and agree that the ease of the strategy we provide here in addition to its high efficiency will make it the method of choice for CRISPR/Cas9 mediated fluorescent protein KIs in organisms.

As the paper was under review we were able to generate two additional KI lines. mScarletmapre1b, Mapre1b (EB1) is a protein that is known to bind to the positive ends of microtubules and we also generated a second version of the cbx1b KI using the newly engineered mGreenLantern as a fluorescent reporter (mGL-cbx1b). We have integrated this new data in the manuscript text, Figures and Supplementary files accordingly.

The data presented is very convincing and has some beautiful images, however the following points could be clarified and/or supported with data:Diagram in S1 represents the homology arms as very long compared to the cargo. Please fix the diagram for clarity.

We thank the reviewer for his/her comment and have modified Figure S1 as suggested.

The need for Cas9-mSA is not demonstrated. For example, an experiment demonstrating the difference in activity between Cas9 alone and Cas9 fused to mSA using short homology. This could be done simply in F0 fish.

We thank the reviewer for his/her comment and have followed this suggestion.

We have done new rounds of F0 injections to address the points raised by the reviewer(s) using Cas9-mSA (Nt SV40NLS + Ct NucleoplasminNLS) and Cas9 (Nt SV40NLS + Ct NucleoplasminNLS) without mSA and a second Cas9 (Ct SV40NLS) without mSA (from the Schier lab, Gagnon et al., 2014). This new data is presented in Table S7. The results show that Cas9-mSA (Nt SV40NLS + Ct NucleoplasminNLS) and Cas9 (Nt SV40NLS + Ct NucleoplasminNLS) without mSA show comparable lethality rates and mosaic KI efficiency in F0s, with Cas9-mSA having slightly higher rates in both. In addition, a second Cas9, i.e. Cas9 (Ct SV40NLS) without mSA (from the Schier lab) shows a lower lethality rate and a lower mosaic KI efficiency in F0s compared to the experiments in which we used Cas9 (Nt SV40NLS + Ct NucleoplasminNLS) or Cas9-mSA (Nt SV40NLS + Ct NucleoplasminNLS).

While the paper was under review we were able to generate a new KI line mGL-cbx1b utilizing Cas9 (Ct SV40NLS) without mSA, albeit with slightly lower efficiency of F0 targeting and germline transmission (updated Table 1 and Table S7). Combined, these results indicate that the strategy we employ also works, in principle, using a Cas9 without mSA. As we have obtained higher efficiencies and importantly, done the validation, confirming single-copy integration specifically in the *locus* of interest, using Cas9-mSA, we recommend using Cas9mSA.

All this new information is added to the updated manuscript.

Data presented in S2 (A, B, C) is very difficult to see and interpret. Please consider revising the figure to enable the font to be read and the data to be interpreted. More description of the conclusions from these experiments would also help the readers.

We thank the reviewer for his/her comment. We have split Figure S2 into Figure S2 and S3 and have increased the font and size of the panels as suggested. In addition, we have augmented the figure legend of S2 to more clearly state the conclusions obtained from this figure as the reviewer suggests.

The manuscript is nicely written and does not overstate the results. The authors were very clear about previous work that led them to the current methodology. This was nicely referenced. The paper also nicely outlines homology arm design.Reviewer #1 (Significance (Required)):As alluded to above the individual aspects of the methodology are well documented in other studies, however, the real significance of the paper is the combination of these methodologies to achieve high rates of precise targeted integration and ease of cargo template production. I predict that this will become the method of choice for targeted integration in model organisms and will be highly referenced. Our laboratory has published similar rates of targeted integration using plasmid templates, but given the ease of production, I look forward to trying and possibly switching to this technology.

We thank the reviewer for his/her comments and are looking forward to his/her lab members trying this approach.

Referee Cross-commentingI agree, but selection is part of the protocol. Their F0 numbers are very high, making selection maybe not as important? I do agree that their numbers are not higher than previous work, but the ease of what they are doing streamlines design. Other advantages include not having a vector and single copy integration.

We agree with Reviewer 1. In addition, we would like to point out that we present 3 measures of efficiency: 1 – F0 mosaic targeting 2 – F0 germline transmission rate 3 – germline transmission efficiency (% of positive F1 embryos per single transmitting F0). All three measures are comparable to highly efficient ones previously reported for medaka (the model organism we are using) (Gutierrez-Triana et al. 2018) and all three are on par (Wierson et al., 2020) or better (Levic et al., 2021) than ones published in zebrafish. As the reviewer acknowledges, a key additional benefit of this approach is its simplicity of implementation, combined with ease of targeting/validation and single copy integration.

I think all three reviewers are in agreement that the utility of Cas9-mSA needs to be addressed. The authors previous paper looked at capping templates with biotin, but this would be an easy control to examine in the current study too, +/- biotin. Wierson et al. showed there does not seem to be a difference between long and short homology arms, so I am not sure how important this is to address here.

We thank the reviewer for his/her comment. We want to point out that the primary reason for using biotin is preventing donor construct concatemerization as previously shown in medaka (Gutierrez-Triana et al. 2018). That said, we have performed new F0 injections with and without biotin (Table S7) and results indicate that there is no clear difference in mosaic KI efficiency in F0s between them.

Regarding the utility of Cas9-mSA, please see our response to reviewer 1 earlier, who raised this point, too.

The efficiency we report using short homology arms is very comparable to a highly efficient KI method previously shown in medaka using long homology arms (Gutierrez-Triana et al., 2018) and is on par (Wierson et al., 2020) or better than recently reported KIs (Levic et al., 2021) in zebrafish. Given the ease of generation of KI constructs (single PCR), the ease of validating single insertions (by sanger sequencing spanning the entire *locus*) and the added benefit of cost reduction of our approach we believe it will be widely preferred and utilized.

Reviewer #2 (Evidence, reproducibility and clarity (Required)):Seleit et al. utilize CRISPR/Cas9-generated Double Strand Breaks (DSBs) to stimulate Homology Directed Repair (HDR) that results in the insertion of sequences encoding reporter proteins into the reading frames of selected genes. They generate many original lines in zebrafish that will be useful to others in the research community. The work is well designed and well performed, the data is clear, the expression patterns are often beautiful, and the lines they create are likely to be useful.

We thank the reviewer for his/her comment. We would like to point out that we are working with Oryzias latipes (medaka), not zebrafish, which is important when evaluating and comparing our results with previous published approaches, see below.

The authors claim they are introducing an improved method for accomplishing HDR. This claim is not well substantiated, and the advantage of their method over other published approaches is not demonstrated.

We would like to answer this general criticism by citing reviewer 1” As alluded to above the individual aspects of the methodology are well documented in other studies, however, the real significance of the paper is the combination of these methodologies to achieve high rates of precise targeted integration and ease of cargo template production (no cloning required to make the repair donor). I predict that this will become the method of choice for targeted integration in model organisms and will be highly referenced.” We share this view of reviewer 1.

The authors utilize an approach in zebrafish that has been shown to have some promise in mice and in tissue culture cells.

We report about our experiments in medaka, not zebrafish, see above.

The idea of the approach is to tether donor template DNA to a CRISPR/Cas9 Ribonucleoprotein (RNP) complex so that repair template is present very close to the targeted* locus* at the time a chromosomal DSB is generated.The authors accomplish this by injecting fertilized zebrafish eggs with a cocktail containing mRNA encoding Cas9 linked to monomeric streptavidin (mSA), a synthetic single guide RNA molecule, and linear dsDNA template that is biotinylated at its 5' ends. The linear donor molecules are created by PCR. The ends of these molecules carry about 40bp of homology to the left and right borders of the DSB break in the host *locus*. HDR results in "pasting in" protein-coding sequences into the endogenous coding *locus* so that the introduced sequences are in-frame and expressed from the targeted *locus*.The method of using short homology arms to introduce donor sequences at a CRISPR/Cas9-targeted site has been explored quite well by other researchers in the field, notably Wierson et al., 2020 and Hisano et al. 2015. The gene fusion of Cas9 linked to monomeric streptavidin was developed by the Rossant lab and the method for generating the donor template by PCR was developed in the Wittbrodt lab. So the questions are: what is new here and is the method an advance? The new element is to introduce a combination of Cas9-mSA and biotinylated donor template to drive HDR in the zebrafish. Does the method in fact lead to interactions between Cas9-mSA and donor molecules? Does the method increase efficiency of HDR events? These questions are not answered.

We would like to point out that our method uses PCR donors with short homology arms, no plasmid repair donors, no in vivo linearization, and no long homology arms, as is reported in previous methods. This clearly simplifies the process of repair donor synthesis and KI generation. We also provided strong evidence for the specificity of such donors using Whole Genome Sequencing. In addition, and as we discuss in the manuscript, the KI lines generated represent important new tools for the community.

There are two critical pieces of information that are missing:– First and foremost: the authors fail to perform parallel, necessary control experiments with non-biotinylated donor templates. Thus it is impossible to know whether biotinylation might enhance complex formation with the RNPs and might enhance the frequency of HDR events.

As pointed out by all reviewers, this is indeed an important point that we have now addressed in the revised manuscript. We performed additional injection experiments, summarized in new Table S7. In these experiments, we compared F0 targeting efficiency based on visual screening of fluorescence and using Cas9 either with or without mSA and in addition, testing the effect of using biotinylated vs non-biotinylated primers. From these results, we conclude that Cas9-mSA (Nt SV40NLS + Ct NucleoplasminNLS) and Cas9 (Nt SV40NLS + Ct NucleoplasminNLS) without mSA show comparable lethality rates and mosaic KI efficiency in F0s, with Cas9-mSA (Nt SV40NLS + Ct NucleoplasminNLS) having slightly higher rates in both.

Results also indicate that the targeting efficiency in F0 using Cas9-mSA (Nt SV40NLS + Ct NucleoplasminNLS) with and without biotinylated primers is comparable. As the purpose of biotinylated primers has been shown previously (Gutierrez-Triana et al., 2018) to be the prevention of concatemerization of exogenous donor DNA, our advocated approach in this manuscript promotes usage of biotinylated primers in order to increase single-copy integration events.

– Second: it is very difficult to decipher the true frequency of HDR events from their work. The authors inject the cocktails in fertilized eggs, and then select F0 embryos that express the reporter protein. Some portion of these (perhaps only those with extensive expression of the reporter?) are raised to adulthood and tested for the ability to transmit a correctly modified *locus* to the next generation. As reported in Table S1, very very few F0 animals are tested for transmission of a modified *locus*. Because of the selection of F0's to be tested, it is hard to know what portion of all the injected F0's have modified genomes that can be transmitted. This strong bias makes it difficult to assess the actual HDR efficiency in donor-injected F0 embryos and to compare the present results with previous published results.

This is an important point as indeed, the approach includes a selection step based on the visual inspection of fluorescence in F0 injected founders. We have adjusted the representation and text accordingly with the aim to further clarify how to estimate efficiencies. To this end, we calculate rates using three complementary readouts:

1. F0 mosaic targeting based on fluorescence screening;

2. Germline transmission rate, pooled: percentage of GLT across tested F0;

3. Germline transmission rate, individual: GLT rate based on positive F1 embryos, per single transmitting F0.

We report that on all three measures, the KI efficiencies we obtain are comparable to highly efficient ones previously reported for medaka (the model organism we are using) (Gutierrez-Triana et al., 2018). In addition, all three measures are on par (Wierson et al., 2020) or better (Levic et al., 2021) than ones published in zebrafish. The added advantage is the simplicity of our approach, its ease of use, ease of validation and precise single copy integration.

We believe that pre-selection is an important aspect of the time and resources saving strategy we report here. Pre-selection is also widely reported and used in previous literature on the subject when reporting on efficiency (for instance, Wierson et al., 2020, Levic et al., 2021, Gutierrez-Triana et al. 2018). But in a hypothetical in which one could not pre-select, the approach we report here would still be extremely useful as the KI efficiencies in F0 and the germline transmission rates are quite high. If a researcher were to inject 100 embryos and have an early lethality of around 40% this leaves him/her with 60 embryos. The researcher can grow all 60 embryos to adulthood and then perform fin-clips on all 60 to identify positive carriers of the insertion (with an average KI efficiency of around 20%, close to what we report, this would be 12 fish). The researcher can then outcross the 12 positive fishes to the WT strain and screen for founders. While the effort put in and resources used are substantially higher than with pre-selection, the method is guaranteed to work.

Minor PointsMain TextLine 127:.…or a mosaic insertion of very low frequency.This seems to be incorrect. Because the WGS is performed on F1 embryos, even if a mosaic insertion occurred with very low frequency in the F0 animals, it would be heterozygous in F1 progeny, but not mosaic.

We thank the reviewer for his/her comment. The reviewer is correct in stating that any insertion will be heterozygote in F1. However, since we pool F1 embryos from the same founder for the F1 deep sequencing it is possible that not all embryos show the same insertion. We have changed the phrasing to ‘rare’ instead of ‘mosaic’ to clarify this point.

Line 365: Should this be Figure 4B-B’ but not 4C-C’?

We thank the reviewer for his/her comment and have corrected this error.

Figure 1 legendLine 574:.… the PCR amplified donor plasmid containing short homology arms.… It should be donor "fragment" not “plasmid".

We thank the reviewer for his/her comment and have corrected this error.

Figure 4 legendLine 657: beginning at 1200 minutes rather than 20:00h.

We have modified the text accordingly.

Supplemental Figure 1 legend (A)? (B)? or one panel?

We have modified the text accordingly.

Supplemental Figure 2 legendThe explanation is not enough to understand the supplemental figure 2.The WGS analysis revealed targeted integration of fluorescent markers, but the reason for a single copy insertion is not clearly explained.

We have split Figure S2 into Figure S2 and S3 and have increased the font and size of the panels as suggested. In addition, we have augmented the figure legend of S2 to outline the evidence for single copy insertion more clearly. Please note Figure S2 is just a visualization using IGV (Interactive Genome Viewer), the details of the bioinformatic tools and steps used to arrive at our conclusions are included in the Materials and methods section: if the donor construct integrated elsewhere in the genome we expect to see mapped reads of the fluorescent reporter sequence overlapping with other endogenous *loci*. If concatemerization occurs at one *locus*, we expect to see reads of the fluorescent reporter repeated in tandem with short homology arms in between. We do not see evidence for either. In addition, Sanger sequencing confirms single copy integration.

Please explain the turquoise bars in A and in B and the dark grey bars in C that are mapped away from predicted integration sites.

As we state stated in the figure legend the turquoise bars represent inter-chromosomal reads that span both the eGFP sequence and the endogenous cbx1b *locus*_._ Regarding the fact that some are mapped slightly away from the predicted integration site: it all depends on where the sequencing read starts. If a fluorescent reporter sequence read happens to be at the very end of the fluorescent reporter sequence we expect that it maps slightly away from the predicted insertion site (which is the beginning of the FP sequence).

What are orange bars?

Orange/red bars are anomalous mappings that either deviate from the expected insert size or paired-end mapping orientation. We have added this in the figure legend.

What are dark grey bars in A and B?

There are no dark grey bars in B.

As per information from IGV: dark grey bars in A represent “unpaired mate, mate not mapped or otherwise unknown status.” The mapping quality of these two reads is 0 and therefore they most likely represent false mapping. We have added this in the figure legend.

What are purple bars in B?

There is one purple bar in B. IGV has flagged this one read as possibly an inter-chromosomal read between chromosome 9 and chromosome 17. The mapping quality of this read is 0 which again means it is in all likelihood a false mapping. We added this in the figure legend.

In D, E, F, each chromatograph shows single peaks, but schematic bars above the chromatographs indicate heterozygotes – is this a conflict?

We are showing sequencing results only for the edited allele in the new Figure S3. The WT allele was also sent to sequencing for all three genes but is not shown here. The schematic bars directly above the chromatographs are simply four colors representing ATGC. They do not indicate heterozygosity. Any difference between the sequencing results and the in-silico in-frame fusion protein is indicated as a black bar within the consensus coverage on the top most panel. Since there are no differences between the in-silico and the sequenced reads this means there are no mismatches. We have modified the figure legend to indicate the location of the consensus sequence more clearly.

Supplemental Figure 5 legendYellow arrowhead in (B): it looks like "otic" vesicle sensory organs but not "optic" vesicle sensory organ.

We thank the reviewer for spotting the mistake. We have corrected it to otic vesicle.

Yellow arrowhead in (G): eGFP expression cannot be seen in the panel.

We thank the reviewer for his/her comment. We have adjusted the location of the Yellow arrowheads in (G) to more clearly represent the vacuolated cells of the notochord. Please note that vacuolated cells have a highly peculiar morphology where the vast majority of the cell body is filled with one large (unlabeled) vacuole. The eGFP signal is therefore weak and located at the thin plasma membrane. We have enhanced the signal in the corresponding panel and have adjusted the text to reflect the weak labeling.

Materials and methodsLine 955 – 996: Authors should mention whether or not each of the biotinylated primers contains phosphorothioate bonds as described in Gutierrez-Triana et al.

We thank the reviewer for his/her comment. We updated the manuscript material and methods, and file S1, to clearly state that the biotinylated primers do not contain any additional modifications.

Line 1004 – 1006: should describe concentration of each component in fin-clip buffer, rather than volume of each stock solution.

We thank the reviewer for his/her comment. We have adjusted the text accordingly.

Reviewer #2 (Significance (Required)):In sum, the zebrafish lines and reagents provided here will be highly useful to the community.In contrast the assertion that the authors have developed a new and improved method for achieving Homology-Directed Repair in zebrafish is not well-supported by the experiments.

Our work was done in medaka, and we agree, these lines and the ones that will be generated with this approach will prove invaluable new tools for the medaka community.

As above, we would agree with reviewer 1 ” As alluded to above the individual aspects of the methodology are well documented in other studies, however, the real significance of the paper is the combination of these methodologies to achieve high rates of precise targeted integration and ease of cargo template production. I predict that this will become the method of choice for targeted integration in model organisms and will be highly referenced.”

Referee Cross-commentingI still think there is insufficient indication that the efficiency they achieve is any different from what anyone else has achieved in the past. all data boils down to Table S1 which tests very few animals, each of which were pre-selected for expression. what would happen if an investigator could not pre-select? I don't think they establish efficiency at all.

We believe that pre-selection is an important aspect of the time and resources saving strategy we report here. Pre-selection is also widely reported and used in previous literature on the subject when reporting on efficiency (for instance, Wierson et al., 2020, Levic et al., 2021, Gutierrez-Triana et al. 2018). But in a hypothetical in which one could not pre-select, the approach we report here would still be extremely useful as the KI efficiencies in F0 and the germline transmission rates are quite high. If a researcher were to inject 100 embryos and have an early lethality of around 40% this leaves him/her with 60 embryos. The researcher can grow all 60 embryos to adulthood and then perform fin-clips on all 60 to identify positive carriers of the insertion (with an average KI efficiency of around 20%, very close to what we report, this would be 12 fish). The researcher can then outcross the 12 positive fishes to the WT strain and screen for founders. While the effort put in and resources used are substantially higher than with pre-selection, the method will still work.

Reviewer #3 (Evidence, reproducibility and clarity (Required)):In the article "Endogenous protein tagging in medaka using a simplified CRISPR/Cas9 knock-in approach" Seleit et al. describe a knock-in method for efficient tagging of endogenous proteins by CRSPR-mediated homologous recombination. The protocol takes advantage of previous advances by combining fluorescent reporters flanked by biotinylated short homology arms together with streptavidin-Cas9. The authors have used this approach to tag six different* loci *in medaka, generating a useful collection of reporter lines (particularly mScarlet-Pcna), initially characterized in this work. The main claims of the article are on the advantages and efficiency of the developed knock-in method and on the added value provided by the lines generated.Overall, the authors provided enough evidence to support the main conclusions of the article. The work is technically sound and I would only request a few additional controls indicated below. The manuscript is well written, the topic is well introduced, the relevant literature referenced, and the data in general terms nicely presented (see minor comment for a few exceptions). Particularly useful are the descriptions of the lines generated (and yet see again minor comments for a few exceptions) and, in particular, the accurate report of the knock-in protocol, which will facilitate other groups to replicate the method.Major Comments:1. Beyond previous knock-in strategies in medaka focused in blocking the 5' ends of the donor molecules (Gutierrez-Triana et al., 2018), the authors are now combining in medaka methods adapted from Gu et al. 2018 (i.e. linking donor to the Cas9 via a streptavidin-biotin system) together with short homology arms (Wierson et al. 2020, Levic et al. 2021). Although the strategy seems extremely effective, the current description does not allow to deduce which of the modifications of the protocol had a bigger impact on its efficiency. Understanding this aspect may be important for further methodological refinements. Particularly, it is difficult to know if the 5' biotin block of the template is by itself sufficient to achieve the high % of integration or if it requires the interaction with the Streptavidin-tagged Cas9. Although it is likely that this interaction is relevant to increase targeting efficiency, this has not been formally demonstrated in the medaka context. A few control injections using regular Cas9, could resolve this issue in F0 embryos.

We thank the reviewer for his/her comment. We have performed the control experiments suggested by the reviewer using Cas9-mSA (Nt SV40NLS + Ct NucleoplasminNLS) and Cas9 (Nt SV40NLS + Ct NucleoplasminNLS) without mSA. In addition, we have tested another Cas9 (Ct SV40NLS), from the Schier lab that is widely used in the teleost community. This new data is presented in Table S7.

The results show that Cas9-mSA (Nt SV40NLS + Ct NucleoplasminNLS) and Cas9 (Nt SV40NLS + Ct NucleoplasminNLS) without mSA show comparable lethality rates and mosaic KI efficiency in F0s, with Cas9-mSA (Nt SV40NLS + Ct NucleoplasminNLS) having slightly higher rates in both. In addition, a second Cas9 without mSA, i.e. Cas9 (Ct SV40NLS) from the Schier lab shows a lower lethality rate and a lower mosaic KI efficiency in F0s compared the experiments in which we used Cas9 (Nt SV40NLS + Ct NucleoplasminNLS) or Cas9-mSA (Nt SV40NLS + Ct NucleoplasminNLS).

As we have obtained highest efficiencies with Cas9-mSA (Nt SV40NLS + Ct NucleoplasminNLS) and importantly, have done the validation confirming single-copy integration specifically in the *locus* of interest using Cas9-mSA (Nt SV40NLS + Ct NucleoplasminNLS), it is the recommended approach put forward in our manuscript. We have updated the manuscript with this new data.

2. Related to the previous point: Although the authors already included a control for the presence of the homology arms, additional controls comparing biotinylated vs unbiotinylated templates will also be informative.

We have performed the control experiments suggested by the reviewer and they are presented in the new Table S7. Results indicate that the targeting efficiency in F0 using Cas9-mSA (Nt SV40NLS + Ct NucleoplasminNLS) with and without biotinylated primers is comparable. As the function of biotinylated primers has been shown previously (Gutierrez-Triana et al., 2018) to be the prevention of concatemerization of exogenous donor DNA, our advocated approach in this manuscript promotes usage of biotinylated primers in order to increase single-copy integration events.

3. In general terms the observations are sufficiently replicated, and the number of embryos observed are properly indicated in the tables and figure legends. Statistic analyses are not required for this work in principle. However, the authors should clarify whether the representation of the mScarlet-PCNA distribution vs time (Figure 4 and Supplementary Figure 8) corresponds to the average plot of 9 epithelial nuclei recorded independently or is a representative individual track.

All data presented in Figure 4 represent measurements from a single cell. In the newly numbered Figure S9 we present individual measurements of mScarlet-Pcna intensity from 9 epithelial cells, in addition to the average (trend). In the newly numbered Figure S10 we present data from the single cell shown in Figure 4A. A similar analysis was done on all 8 other cells and we show the results in the new Figure S11.

Given the importance of this line as a tool to follow in vivo the cell cycle progression, it would be necessary to show how variable these measurements are, either by plotting the average distribution + SD, or by showing several individual recordings.

We plot the individual measurements of intensity over time in Figure S9 and we also plot their average combined. Cells progressing into S phase show an increase of endogenous Pcna levels, peak pixel intensity distribution marks the transition from S/G2 (seen as bright speckles in the time-series) and a sharp drop in endogenous nuclear Pcna levels marks the beginning of M phase (Figure S9). In addition, we have now included individual measurements of pixel intensity distribution over time from all 8 other cells marking the transition from S/G2 in new Figure S11 as the reviewer suggests.

Minor points:1. Although in general the knock-in lines are sufficiently described in this work; a few co-staining experiments will help to explore the universality of the tools generated. For example, a simple co-staining with DAPI may help to deduce if the line eGFP-cbx1b is a general nuclei marker.

We have performed a double staining as requested. Results show the overlap between Hoechst and the anti-eGFP antibody in the eGFP-cbx1b line. The results are presented in new Figure S5.

Similarly, a co-labelling with an endosomal marker would be a nice conformation of the intracellular distribution of the eGFP-Rab11a fusion.

We have confirmed that the fusion protein generated is in-frame. Future work will focus on a detailed characterization of the line.

2. Figures and videos of medaka tissues (figures 1-3, Supplementary figures 3-6, and Supplementary videos 1-7) are in general poorly labelled. In order to facilitate the interpretation of the data through self-explanatory figures, I would recommend adding a few abbreviations indicating key anatomical landmarks in each panel. It would also help including a legend stating which structure is included in each panel, as it has been done in Figure 3.

We thank the reviewer for his/her comment. We have added the suggested annotations in Figure 1, 2 and 3 and have modified the figure legends as requested. And we have also added the suggested annotations in Supplementary Figures 4-8 and modified the legends as requested.

3. The manuscript includes some unnecessary claims of novelty that do not add much to the interpretation of the text and are in some cases questionable [e.g. page 10; "first reported fusion-protein for a cadherin"; or "first reported endogenous ubiquitous nuclear marker in teleosts"; page 8 "provide first evidence that the mScarlet-pcna line recapitulates known dynamics of Pcna within the nucleus]. I would suggest to remove them.

We thank the reviewer for his/her comment. We have removed references to ‘first’ in the two instances outlined. We noted that our statement "provide first evidence that the mScarlet-pcna line recapitulates known dynamics of Pcna within the nucleus” was misleading and have corrected it to read "provide initial evidence that the mScarlet-pcna line recapitulates known dynamics of Pcna within the nucleus”.

4. Text accompanying Supplementary figure S2, as well as the figure itself, are difficult to follow. Increasing the fonts size and the resolution of the images in panels A-C will improve their interpretation. In the text, it will also help referring to the individual panels instead of the whole figure. e.g. "identify paired-end eGFP reads anchored to the endogenous cbx1b *locus*.… (Figure S2)"…should be Figure S2A.

We thank the reviewer for his/her comment and have followed all the suggestions.

5. Figure legend of S5: the otic capsule (yellow arrowhead in B) is wrongly indicated as "optic vesicle”.

We thank the reviewer for spotting the mistake. We have adjusted the text accordingly.

Reviewer #3 (Significance (Required)):While the work is technically very solid and the described method seems valuable, I still have mixed feelings on its potential impact in the field. The fact that the technology developed is a refinement of published methods (Particularly Gu et al. 2018, Gutierrez-Triana et al., 2018, Wierson et al. 2020) reduces the novelty of the study. That being said, I think the protocol guarantees a precise and high-efficient integration of the reporters with limited experimental effort, and thus the advance is significant.

We thank the reviewer for these supporting comments.

The method will be of interest to groups working in teleost genetics (such as this referee), mainly to those working with medaka, which may take advantage of the helpful knock-in lines generated. In that regard, the impact of the work could be greater if a few comparative experiments, testing the protocol applicability in other teleost models, were included. This is particularly the case for zebrafish, given the size of the research community working with this model.

We agree with the reviewer that future work should address the potential of this approach in other organisms including zebrafish, but this is beyond the scope of this paper. Our initial results from pilot injections in a number of other organisms suggest that the approach is both functional and efficient, as such we are expecting its adoption both in the teleost field and most likely beyond.

Referee Cross-commentingI do agree with the last comment by reviewer #1. Selection is not as important. Being practical, the current protocol guarantees an efficiency level that is already high enough. Improving this aspect is likely not a key methodological demand.